# Vision Function Layer in Multimodal LLMs

**Cheng Shi**[1,2]*, **Yizhou Yu**[2], **Sibei Yang**[1]†
[1]Sun Yat-sen University, [2]School of Computing and Data Science, The University of Hong Kong
shicheng2025@connect.hku.hk, yizhouy@acm.org, yangsb3@mail.sysu.edu.cn
https://github.com/ChengShiest/Vision-Function-Layer

## Abstract

This study identifies that visual-related functional decoding is distributed across different decoder layers in Multimodal Large Language Models (MLLMs). Typically, each function, such as counting, grounding, or OCR recognition, narrows down to two or three layers, which we define as Vision Function Layers (VFL). Additionally, the depth and its order of different VFLs exhibits a consistent pattern across different MLLMs, which is well-aligned with human behaviors (e.g., recognition occurs first, followed by counting, and then grounding). These findings are derived from Visual Token Swapping, our novel analytical framework that modifies targeted KV cache entries to precisely elucidate layer-specific functions during decoding. Furthermore, these insights offer substantial utility in tailoring MLLMs for real-world downstream applications. For instance, when LoRA training is selectively applied to VFLs whose functions align with the training data, VFL-LoRA not only outperform full-LoRA but also prevent out-of-domain function forgetting. Moreover, by analyzing the performance differential on training data when particular VFLs are ablated, VFL-select automatically classifies data by function, enabling highly efficient data selection to directly bolster corresponding capabilities. Consequently, VFL-select surpasses human experts in data selection, and achieves 98% of full-data performance with only 20% of the original dataset. This study delivers deeper comprehension of MLLM visual processing, fostering the creation of more efficient, interpretable, and robust models.

## 1 Introduction

Large language models (LLMs)[2, 3, 8, 17, 47, 55, 57, 59, 60], built on deep transformer layers [61], have become the dominant paradigm in natural language processing, demonstrating remarkable versatility and human-level performance across diverse tasks [16, 45, 53, 56, 69]. Recent studies [11, 38, 44] further reveal that frontier LLMs develop hierarchical internal structures and problem-solving strategies analogous to human cognition, in which simple features are combined into complex representations and tasks are decomposed into sub-components. Understanding these mechanisms has become central to improving interpretability [20, 32], efficiency [11, 76], and driving architectural advances like Mixture of Experts [1, 15, 66].

Extending LLMs, Multimodal Large Language Models (MLLMs) [3, 6, 7, 23, 64] achieve joint text-vision understanding and reasoning by processing integrated visual and textual inputs. Through supervised fine-tuning on vision instruction data, these MLLMs have progressed beyond simple image captioning to address diverse tasks requiring visual perception and understanding. However, despite these remarkable advancements in visual understanding capabilities, the internal workings of these MLLMs—particularly how they process and reason with vision tokens—remain largely

---

*Work done during Cheng's internship at SYSU.
†Corresponding author is Sibei Yang.

39th Conference on Neural Information Processing Systems (NeurIPS 2025).

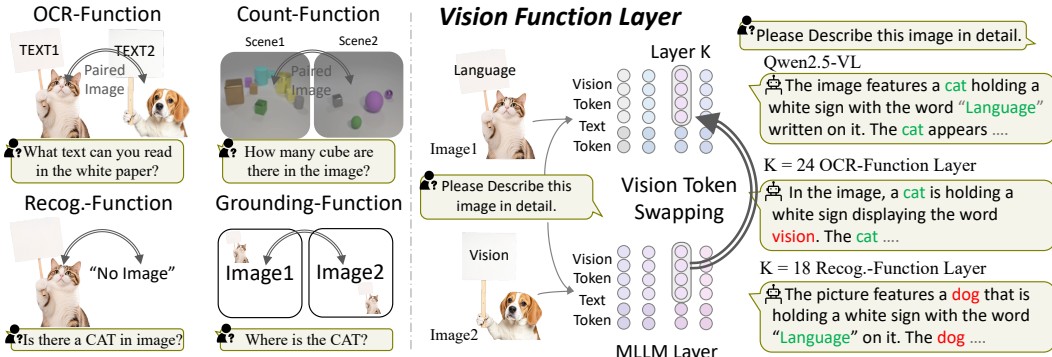

Figure 1: **Overview of the Vision Token Swapping Framework.** Left: Probing specific visual functions using minimally different image pairs and targeted questions. Right: The token swapping mechanism, where vision tokens from a source image replace those of a target image at a specific layer within the MLLM's KV cache during decoding. The example with Qwen2.5-VL demonstrates how swapping at different functional layers (Layer=24 for OCR, Layer=18 for Recognition) directly alters the model's output to reflect the swapped content.

unclear, often characterized as a "black box." The heightened challenge in understanding the internal mechanisms of MLLMs, compared to their text-only counterparts, stems primarily from two aspects:

- *First*, the diversity of vision-language tasks [22, 30, 35, 36, 68] tackled by MLLMs demands mastery of a wide range of fundamental visual functions, where we define a visual function as a distinct perceptual capability essential for solving a specific category of vision tasks—such as object recognition, counting, text reading, or spatial reasoning—each reflecting a particular type of visual understanding. These atomic visual functions serve as building blocks for more complex and integrative vision-language tasks, where multiple perceptual capabilities are jointly required. For instance, solving a math problem in a diagram may require reading handwritten equations (OCR), counting geometric elements, and reasoning about spatial relationships—each engaging different visual functions in concert. Unlike text-only models that operate in a uniform linguistic space [28, 31], MLLMs must learn to activate and combine heterogeneous visual functions to interpret diverse visual inputs. This multi-faceted requirement adds substantial complexity to their internal mechanisms, making it difficult to pinpoint how specific visual functions are represented, combined, and aligned with textual reasoning.

- *Second*, unlike text-only LLMs, which have largely adopted standardized architectures, MLLMs still present diverse designs, especially in their visual branches. They employ different types of visual encoders [10, 39, 43, 63] and various connector modules [6, 25, 29, 58] to integrate vision tokens with text tokens. This architectural variability further complicates the understanding of their internal mechanisms — akin to assuming that different species, each sensing the world through distinct systems, would nonetheless process these signals through identical neural pathways [54].

While recent MLLM studies [4, 9, 37, 71] have explored token importance and cross-modal interactions, they largely overlook how diverse visual functions are internally represented and coordinated. A key challenge lies in the lack of diagnostic frameworks that isolate individual functions, as most general-purpose tasks require multiple abilities simultaneously. This limitation hinders targeted analysis and leads to only coarse conclusions (e.g., shallow layers extract visual features while deeper perform reasoning), leaving core questions about MLLMs' internal visual mechamisn unanswered.

*To address this challenge, we examine the layer-wise functional roles of vision tokens within the LLM backbone of MLLMs, aiming to understand how different layers contribute to the realization of specific visual functions.* We propose a two-level, step-by-step evaluation framework: single-function evaluation via visual tokens swapping and multi-function evaluation via visual tokens drop. Our single-function evaluation assesses the functional roles of visual representations across layers by swapping visual tokens between pairs of images differing solely in one visual function. This controlled perturbation reveals how layer-specific visual features contribute to the model's output, as illustrated in Fig. 1. Building on this, we extend our evaluation to multi-function general benchmarks [12, 22, 24, 30, 35, 68]. In these more complex settings, where precisely designing pairs of images that differ by only a single visual function for token swapping is challenging, we

instead employ a token dropping strategy. By analyzing performance degradation after dropping visual tokens from different layers, we can identify the importance of specific layers for various tasks within these benchmarks and subsequently infer the critical visual functions these layers support.

We comprehensively test across a diverse range of vision functions and MLLM architectures, leading to the surprising discovery of a consistent internal MLLM mechanism. This mechanism proves broadly applicable, from early MLLM iterations like the LLaVA series [23, 29] to recent models such as the Qwen series [3, 62]. Our key findings are as follows:

- **MLLMs feature Vision Function Layers,** where specific visual functions are executed within remarkably narrow layer blocks (typically 2-3 layers). Qwen-2.5-VL, for example, restricts count-function to layers 14-16 and OCR to layers 22-24. This division of labor is sharply defined: these functional layers operate with exclusivity, and other layers contribute negligibly to these specific tasks.

- **Vision Function Layers exhibit a consistent arrangement in diverse MLLMs,** where recognition typically occurs earliest, followed by counting in middle layers, then grounding, and finally OCR in later layers. This observed sequence holds true across MLLM generations (from LLaVA-v1.5 to Qwen2.5-VL) and scales (3B to 70B).

- **Vision Function Layers are redundant within MLLMs.** For tasks like ScienceQA and MMMU, which do not rely on function-specific layers, MLLMs often maintain or even improve performance when redundant Vision Function Layers, typically constituting over half of the model's depth, are omitted.

This mechanisms shed light on the "black-box" nature of MLLMs, offering explanations for previously puzzling phenomena and the diverse behaviors of MLLMs across applications. We believe these insights are fundamental to numerous MLLM applications, such as guiding the development of function-layer-targeted parameter-efficient fine-tuning strategies and enabling more principled vision instruction data selection based on active function layers.

In summary, our key contributions are as follows:

1. We propose a novel evaluation framework centered on visual token swapping and dropping. This framework operates by replacing carefully designed paired image data to precisely locate the functional layer of different tasks, providing a unique methodology for analyzing MLLMs behavior.

2. We provide comprehensive findings obtained through the application of our framework to a wide range of MLLMs and various visual tasks. These evaluations reveal a consistent layer-wise functional arrangement across different model families, successive versions, and model sizes, with specific visual functions consistently mapped to narrow, dedicated layer blocks.

3. We demonstrate the profound practical utility of our insights, showcasing that functional-layer targeting enables: (a) vision-function LoRA, using only one-third the tunable parameters of full LoRA, matches its in-domain performance while boosting out-of-domain generalization; (b) data selection strategies surpassing human experts under identical budget constraints; and (c) achieving 98% of full-data performance with merely 20% of the data.

## 2 Vision Function Layer

### 2.1 Preliminaries on Multi-Model Large Language Models

Let $\mathbf{I} \in \mathbb{R}^{H \times W \times C}$ denote the input image and $\mathbf{T}$ the tokenized text prompt. MLLMs first employ a vision encoder $\mathcal{E}_v$ to map $\mathbf{I}$ into $N_v$ dense embeddings $\mathbf{V}$. These raw visual tokens are then projected into the language model's embedding space by a lightweight connector $\mathcal{P}$, yielding aligned vision embeddings $\mathbf{U}$. In parallel, the text sequence $\mathbf{T}$ is embedded into $\mathbf{W}$. MLLMs concatenate vision and text embeddings along token dimension to form the joint input $[\mathbf{U}; \mathbf{W}]$, which is then processed by $L$ successive Transformer layers $\{\Phi^{(l)}\}_{l=1:L}$. In the prefilling stage, in which the model builds up its multimodal context before any token is generated as follows:

$$[\mathbf{U}^{(l)}; \mathbf{W}^{(l)}] = \Phi^{(l)}([\mathbf{U}^{(l-1)}; \mathbf{W}^{(l-1)}]). \tag{1}$$

During autoregressive decoding, each generated token continuously gathers information from the vision and text token representations at every layer, thereby enabling dynamic cross-modal interaction as follows:

$$P(\mathbf{y} \mid \mathbf{U}, \mathbf{W}) = \prod_{t=1}^{N_{\text{gen}}} P(y_t \mid y_{<t}, \mathbf{U}, \mathbf{W}), \tag{2}$$

where $N_{\text{gen}}$ denotes the total number of tokens generated and $\mathbf{y} = (y_1, y_2, \ldots, y_{N_{\text{gen}}})$ denotes the sequence of output tokens, predicted by projecting the final layer representation through a linear layer followed by softmax over the vocabulary.

## 2.2 Decoding with Vision Token Swapping and Dropping

In this work, we systematically probe the layer-wise vision representations $\mathbf{U}^{(l)}$ to quantify their individual contributions to the predicted token. To probe the function role of vision tokens at different layers, we first introduce *Vision Token Swapping*: at layer $k$, we replace the original vision tokens $\mathbf{U}^{(k)}$ with an alternative set $\widetilde{\mathbf{U}}^{(k)}$, while keeping all other layers unchanged. The resulting decoding probability becomes:

$$P_{\text{swap}}(\mathbf{y} \mid \mathbf{U}^{(\neq k)}, \widetilde{\mathbf{U}}^{(k)}, \mathbf{W}) = \prod_{t=1}^{N_{\text{gen}}} P(y_t \mid y_{<t}, \mathbf{U}^{(\neq k)}, \widetilde{\mathbf{U}}^{(k)}, \mathbf{W}), \tag{3}$$

where $\widetilde{\mathbf{U}}^{(k)}$ can be substituted with vision tokens from any other image, or even replaced with *NULL* tokens. By carefully designing $\widetilde{\mathbf{U}}^{(k)}$, we can assess how vision tokens at layer $k$ influence the generated output, thereby revealing their causal contribution to multimodal decoding.

Next, we consider an alternative probing method for scenarios where generating a specific alternative set of vision tokens $\widetilde{\mathbf{U}}^{(k)}$ is not feasible or desired. In this approach, instead of swapping tokens, we directly drop them. We observed that merely nullifying the vision tokens at a single layer $k$ (i.e. removing $\mathbf{U}^{(k)}$ without replacement and without new information from an alternative source $\widetilde{\mathbf{U}}^{(k)}$) often yields changes in the output that are too subtle to be clearly indicative. To elicit a more discernible impact and assess the cumulative importance of vision information processed up to a certain depth, we adopt a strategy of progressively dropping all vision tokens from a given layer $k$ onwards, named as *Vision Token Dropping*. The decoding probability when all vision tokens from layer $k$ onwards are dropped is formulated as:

$$P_{\text{drop}}(\mathbf{y} \mid \mathbf{U}^{(<k)}, \mathbf{W}) = \prod_{t=1}^{N_{\text{gen}}} P(y_t \mid y_{<t}, \mathbf{U}^{(<k)}, \mathbf{W}), \tag{4}$$

where $\mathbf{U}^{(<k)}$ denotes the visual tokens propagated up to layer $k$. In the subsequent experiments, we adopt Equ. 3 whenever a valid replacement $\widetilde{\mathbf{U}}^{(k)}$ is available; otherwise, we adopt Equ. 4.

## 2.3 Targeting Vision Function Layer by Vision Token Swapping and Dropping

**Experiment Setting.** To precisely identify the layers for key visual functions within MLLMs, we employ our Vision Token Swapping methodology, which measures the "change rate" in the outputs after token swapping. We construct dedicated paired image datasets for four key visual functions: Optical Character Recognition (OCR), Object Counting (Count), Object Recognition (Recognition), and Object Grounding (Grounding), as exemplified in Fig. 1. Each image pair is meticulously designed to isolate a single visual attribute, ensuring minimal differences between paired images and we random choose one as targe image and another as source image. Specifically:

- **OCR** pairs consist of distinct words (sampled from a deduplicated arXiv corpus [61]) rendered onto visually uniform blank canvases, designed to evaluate the model's capacity for textual information extraction. Change rate is quantified by whether the model's output text changes.

- **Grounding** pairs present identical objects placed at varying random locations within otherwise clean backgrounds, aiming to probe spatial sensitivity. Change rate is the proportion of instances where the Intersection-over-Union (IoU) between the predicted bounding box and the swapping-ground-truth bounding box exceeds 0.5.

- **Counting** pairs, adapted from the CLEVR dataset [19], differ primarily in the quantity of a target object type, with associated queries focused on enumeration. Change rate is computed based on whether the predicted number changes.
- **Recognition** pairs, drawn from COCO [27], contrast images containing a target object (e.g., a cat) with blank canvases; queries ask whether this target object is present. The change rate is the proportion of "No" predictions after token swapping.

These experiments primarily utilize the Qwen-2.5-VL-7B model [64] which contains 28 layers, and we have observed similar functional localization patterns across other MLLMs.

**Experiment Results.** Our interventions reveal that specific visual functions are handled in remarkably narrow Vision Function Layers within MLLMs, as illustrated for Qwen-2.5-VL-7B in Fig. 2. "Results Change Rate (%)" is detailed in experiment settings. Collectively, these results highlight a clear hierarchical processing strategy within the MLLM, with distinct layers specializing in different visual functions, from foundational identity cues in early layers to complex OCR-related textual cues in deeper layers.

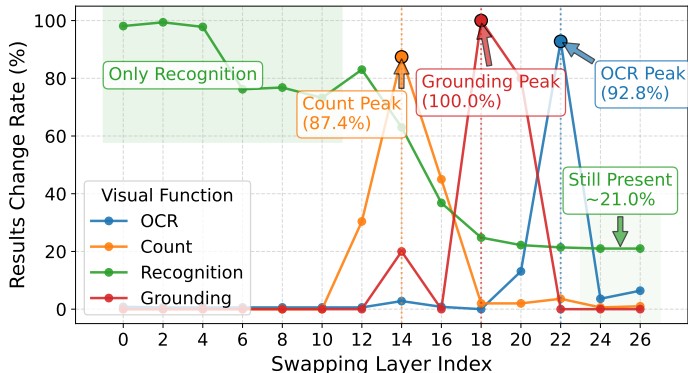

**Recogn.** shows high sensitivity in early layers (0-10), with basic visual features established early, though some effect persists deeper. **Counting** peaks around layer 12 (87.4%), and **Grounding** peaks at layer 18 (100.0%), suggesting mid-layer processing for spatial and numerical reasoning. **OCR** peaks sharply in later layers (layer 22, 92.8%), indicating that visual-linguistic representations are finalized at last stages.

Figure 2: **Vision Function Layer emerges in MLLMs.** A higher Results Change Rate (%) quantifies the involvement of the corresponding layer in processing the corresponding vision function. Functional peaks are sharply localized to specific layers, while other layers contribute negligibly. An exception is recognition function, which peaks in early layers but exhibits distributed influence across almost all layers.

This layer-wise order reflects an MLLM strategy of progressive abstraction: starting with coarse object identification, advancing to conceptual understanding, and culminating in highly specialized representations for tasks like OCR.

## 2.4 Targeting Vision Function Layer by Vision Token Dropping

**Experiment Setting.** To assess whether the layer-specific dependencies observed with our curated paired datasets generalize to broader visual question answering (VQA) contexts, we extend our investigation to general-purpose VQA benchmarks. In these experiments, we evaluate the impact of *progressively dropping vision tokens* as defined in Equ. 4. We test models such as LLaVA-v1.5 (7B, 13B) [23, 29] and Qwen2.5-VL (3B, 7B) [3, 64] across a suite of benchmarks including SQA-I [33], MMMU [68], POPE [22], SEED [24], CVBench [58], TextVQA [52], OCR [30], and ChartQA [35]. Performance is measured using the standard accuracy metrics pertinent to each benchmark. Tab. 4 presents the detailed results, showing performance when all vision layers are used (baseline) versus when an increasing number of final layers are omitted.

**Experiment Results.** Across different models and scales (see Tab. 4), our conclusions are:

1. **Different MLLMs show remarkable consistency in the hierarchical arrangement of the vision function layer.** This hierarchical order strongly corroborates findings from our earlier paired-image swapping experiments. Both experimental approaches reveal that MLLMs process visual information hierarchically, where OCR capabilities decline first (with both

| Method | General & Knowledge | | | Recognition | | Spatial | OCR & Chart | | |
|---|---|---|---|---|---|---|---|---|---|
| | SQA-I [33] | MMMU [68] | MME [12] | POPE [22] | SEED [24] | CVBench [58] | TextVQA [52] | OCR [30] | ChartQA [35] |
| LLaVA-v1.5-7B-32-layer | 68.8 | 34.7 | 1455.9 | 86.4 | 67.3 | 56.2 | 47.2 | 33.0 | 22.0 |
| – *drop 8 v-layers* | 68.8 | 34.4 | 1460.6 | 86.4 | 67.3 | 56.4 | 44.8▼ | 31.7▼ | 21.0▼ |
| – *drop 16 v-layers* | 68.7 | 34.3 | 1457.3 | 86.0 | 67.3 | 53.7▼ | 18.1 | 10.1 | 15.9 |
| – *drop 20 v-layers* | 69.0 | 35.2 | 1470.4 | 83.2▼ | 65.8▼ | 43.0 | 13.1 | 3.7 | 13.8 |
| – *drop 24 v-layers* | 65.7▼ | 33.9▼ | 855.5 | 38.1 | 45.6 | 37.2 | 9.7 | 1.9 | 12.8 |
| LLaVA-v1.5-13B-40-layer | 72.7 | 35.4 | 1522.6 | 85.9 | 68.2 | 53.0 | 48.7 | 33.5 | 22.6 |
| – *drop 8 v-layers* | 72.6 | 35.8 | 1528.3 | 86.0 | 68.2 | 53.2 | 44.8▼ | 31.2▼ | 21.1▼ |
| – *drop 16 v-layers* | 72.7 | 35.4 | 1547.0 | 84.9 | 68.0 | 53.2 | 16.6 | 6.7 | 15.8 |
| – *drop 20 v-layers* | 72.2 | 37.2 | 1458.2▼ | 74.9▼ | 65.8▼ | 51.2▼ | 12.0 | 2.2 | 15.0 |
| – *drop 24 v-layers* | 70.1▼ | 34.6▼ | 783.1 | 11.5 | 48.2 | 52.2 | 9.2 | 2.1 | 13.5 |
| Qwen2.5-VL-3B-36-layer | 80.3 | 46.3 | 1530.9 | 87.0 | 74.8 | 72.9 | 77.8 | 77.8 | 83.4 |
| – *drop 4 v-layers* | 80.0 | 46.9 | 1528.1 | 86.9 | 74.9 | 72.9 | 59.3 | 56.4 | 78.6 |
| – *drop 8 v-layers* | 80.1 | 46.5 | 1530.8 | 86.9 | 74.9 | 72.6 | 22.0 | 17.9 | 60.0 |
| – *drop 16 v-layers* | 79.6 | 46.2 | 1400.9▼ | 82.7▼ | 66.2▼ | 56.4 | 12.4 | 2.5 | 13.0 |
| – *drop 20 v-layers* | 76.7▼ | 45.2▼ | 905.1 | 19.4 | 54.1 | 47.2 | 10.9 | 2.3 | 13.2 |
| Qwen2.5-VL-7B-28-layer | 87.2 | 50.7 | 1696.4 | 86.1 | 77.6 | 80.8 | 82.8 | 82.2 | 83.2 |
| – *drop 4 v-layers* | 87.4 | 50.8 | 1693.6 | 86.3 | 77.5 | 81.0 | 74.1 | 76.3 | 82.7 |
| – *drop 8 v-layers* | 87.4 | 50.6 | 1683.1 | 86.2 | 77.5 | 80.6 | 15.3 | 5.5 | 20.5 |
| – *drop 12 v-layers* | 87.2 | 50.2 | 1633.9▼ | 79.5▼ | 74.5▼ | 69.1 | 13.8 | 3.7 | 17.4 |
| – *drop 18 v-layers* | 77.3▼ | 45.8 | 1111.5 | 37.1 | 52.4 | 44.2 | 12.2 | 2.3 | 14.3 |

Table 1: **Vision Token Dropping on General Benchmarks.** $A_\blacktriangledown$ indicates the onset of performance degradation, while $\underline{A}$ highlights significant drops. The results reveal a consistent hierarchical order of vision function layers across diverse MLLMs. Results of other MLLMs are provided in Appendix.

> models starting to lose OCR functionality between layers 4-8), followed by spatial reasoning, and finally, object recognition.
>
> 2. **Many tasks do not necessitate visual tokens from every layer**, and critically, some tasks achieve superior performance when specific, seemingly non-contributory vision function layers are omitted. For example, on the MMMU task, all models achieved their highest performance when some vision tokens were dropped, and the highest increase could be 1.8%.

# 3 Driving Progress in Multimodal LLMs with Vision Function Layer Insights

## 3.1 Vision-Function-LoRA

**Motivation.** Fine-tuning pre-trained MLLMs is widely used to strengthen specific abilities such as spatial reasoning [58, 65]. Due to their large size, PEFT methods like LoRA [18] have become standard. However, LoRA is typically applied uniformly across layers, which is suboptimal: as our analysis (Fig. 2) shows, different visual functions are localized to specific layers and dropping useless function layers can improve the performance. Moreover, task-specific fine-tuning risks degrading general performance through catastrophic forgetting. To address this, we propose Vision-Function LoRA (VFL), a PEFT method that selectively applies LoRA updates only to layers critical for the target visual function(s), thereby enhancing desired skills while preserving overall model capability.

**Experiment Setting.** To evaluate the efficacy and benefits of VFL-LoRA, we focus on enhancing spatial reasoning—a fundamental visual understanding capability where current MLLMs often exhibit deficiencies. It is important to note that, *to evaluate VFL-LoRA's generalizability and the robustness of the identified Vision Function Layers, we directly select the layers with non-zero change rate of count-function from Fig. 2, without any access to the training or test data of the downstream spatial reasoning benchmarks. For example, for Qwen2.5-VL-7B, we use layers 10–17, 20, 21, 22, and 23.*

We utilize the SAT [46] as training dataset, specifically its single-image question-answering tasks probing spatial understanding. Our base architectures are the Qwen2.5-VL models [64]. We benchmark

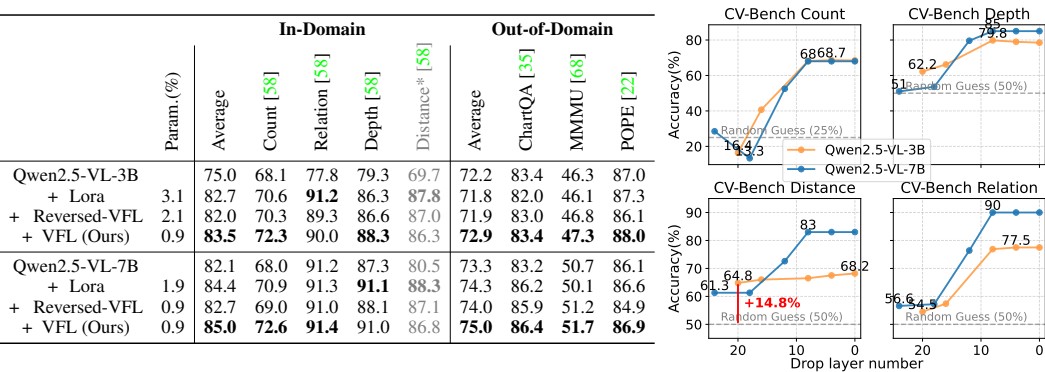

| | Param.(%) | In-Domain | | | | | Out-of-Domain | | | |
|---|---|---|---|---|---|---|---|---|---|---|
| | | Average | Count [58] | Relation [58] | Depth [58] | Distance* [58] | Average | ChartQA [35] | MMMU [68] | POPE [22] |
| Qwen2.5-VL-3B | | 75.0 | 68.1 | 77.8 | 79.3 | 69.7 | 72.2 | 83.4 | 46.3 | 87.0 |
| + Lora | 3.1 | 82.7 | 70.6 | **91.2** | 86.3 | **87.8** | 71.8 | 82.0 | 46.1 | 87.3 |
| + Reversed-VFL | 2.1 | 82.0 | 70.3 | 89.3 | 86.6 | 87.0 | 71.9 | 83.0 | 46.8 | 86.1 |
| + VFL (Ours) | 0.9 | **83.5** | **72.3** | 90.0 | **88.3** | 86.3 | **72.9** | **83.4** | **47.3** | **88.0** |
| Qwen2.5-VL-7B | | 82.1 | 68.0 | 91.2 | 87.3 | 80.5 | 73.3 | 83.2 | 50.7 | 86.1 |
| + Lora | 1.9 | 84.4 | 70.9 | 91.3 | **91.1** | **88.3** | 74.3 | 86.2 | 50.1 | 86.6 |
| + Reversed-VFL | 0.9 | 82.7 | 69.0 | 91.0 | 88.1 | 87.1 | 74.0 | 85.9 | 51.2 | 84.9 |
| + VFL (Ours) | 0.9 | **85.0** | **72.6** | **91.4** | 91.0 | 86.8 | **75.0** | **86.4** | **51.7** | **86.9** |

Figure 3: **VFL-LoRA Efficiency and Diagnostic Analysis on CV-Bench. Left Table:** VFL-LoRA, which trains LoRA adapters exclusively on Count-Function Layers, achieves significant parameter efficiency while maintaining competitive in-domain performance and demonstrating superior out-of-domain generalization across diverse benchmarks. **Right Figure:** Analysis of CV-Bench sub-tasks (Count, Depth, Distance, Relation) using vision token dropping. Results show high visual dependency for Count and Depth sub-tasks, contrasting with strong language priors for Distance and Relation. Attributing its strength in vision-heavy tasks like counting to its focus on Vision Layers, this analysis shows VFL yields clear improvements in this domain, with less impact on language-focused tasks.

VFL-LoRA against two primary baselines: (1) Standard LoRA, where LoRA is applied uniformly across all adaptable layers, and (2) Reversed-VFL, an ablation study where LoRA is applied to layers excluding the count-function layer range. The evaluation is conducted on a comprehensive test set comprising both in-domain spatial reasoning tasks from CV-Bench (which includes sub-tasks like Count, Relation, Depth, and Distance) and a diverse suite of out-of-domain benchmarks (such as ChartQA [35], OCRBench [30], MMMU [68], and POPE [22]) to assess broader generalization. Detailed results are presented in Fig. 3

**Experiment Results.** The performance of VFL-LoRA, primarily benchmark against standard LoRA, is detailed in Fig. 3. Notably, VFL-LoRA achieves a substantial reduction in tunable parameters, requiring nearly 50% fewer (155M vs. 309M for standard LoRA on the tested models). Beyond this increased parameter efficiency, VFL-LoRA attains marginally superior average in-domain accuracy on the CV-Bench spatial reasoning tasks (84.4% vs. 85.0% for standard LoRA), with particular improvements on specific sub-tasks such as CV-Count (72.6% vs. 70.9%). However, standard LoRA showed a lead on CV-Distance (A detailed analysis is provided in Deeper Lock at CV-Bench) More critically, for out-of-domain generalization, VFL-LoRA consistently surpasses standard LoRA, achieving a higher average performance (75.0% vs. 74.3%) across the diverse suite of benchmarks including ChartQA, OCRBench, MMMU, and POPE. In essence, these results indicate that VFL-LoRA not only provides significant parameter savings but also largely maintains or even enhances performance, especially in out-of-domain generalization, compared to the standard LoRA.

**A Deeper Look at CV-Bench.** To further dissect visual dependencies across spatial reasoning tasks in CV-Bench, we apply vision token dropping to Qwen2.5-VL. As shown in Fig. 3, we observe clear task-specific patterns. Performance on Count and Depth drops sharply as more layers are removed, eventually nearing random levels—confirming their strong dependence on processed visual input. In contrast, Distance and Relation remain robust even with heavy layer dropping, suggesting they rely more on language priors and statistical biases rather than detailed visual features. This explains why VFL-LoRA consistently improves count-centric and other perception-heavy tasks, but offers limited gains on tasks like CV-Distance that rely less on the targeted visual functions.

## 3.2 Data Selection through the Lens of Vision Function Layers

**Motivation.** The demand for large-scale instruction datasets to train MLLMs presents significant challenges, as these datasets, while rich in diverse signals, often have heterogeneous quality, making it difficult to determine the specific contribution of individual instances to enhancing distinct model capabilities. This ambiguity complicates efficient training and targeted skill development, thereby necessitating more efficient data selection strategies. To address this, we propose **VFL-Select**, a

| | | General | | | Knowledge | | | OCR & Chart | | | Vision-Centric | |
|---|---|---|---|---|---|---|---|---|---|---|---|---|
| | $Data$ | $MME^P$ | $SEED^I$ | $GQA$ | $SQA^I$ | $MMMU^V$ | $AI2D$ | $ChartQA$ | $OCR$ | $TextVQA$ | $Count$ | $Dist$ |
| Oracle | *665k* | 1476.9 | 67.3 | 63.0 | 86.4 | 34.7 | 62.5 | 22.0 | 33.0 | 47.2 | 34.1 | 43.0 |
| Random | *150k* | 1306.6 | 59.3 | 50.0 | 64.0 | 33.4 | 50.9 | 27.0 | 30.3 | 44.7 | 34.4 | 49.7 |
| | *250k* | 1411.6 | 61.3 | 52.7 | 59.3 | 37.3 | 52.9 | 28.0 | 33.6 | 46.1 | 38.1 | 53.8 |
| | *350k* | 1358.3 | 62.5 | 54.5 | 61.8 | 36.1 | 53.9 | 31.7 | 24.2 | 47.0 | 37.6 | 52.3 |
| | *665k* | 1410.8 | 64.7 | 56.7 | 60.4 | 36.5 | 57.1 | 33.7 | 22.8 | 48.2 | 33.1 | **49.2** |
| Expert [58] | *150k* | 1338.3 | 56.3 | 51.8 | 64.5 | 33.8 | 52.1 | 28.0 | 15.9 | 44.0 | 38.3 | 51.1 |
| | *250k* | 1337.6 | 59.7 | 53.5 | 62.7 | 35.1 | 53.4 | 29.5 | 17.0 | 44.9 | 35.1 | 46.8 |
| | *350k* | 1360.8 | 60.5 | 55.1 | 62.4 | 34.4 | 56.0 | 31.4 | 16.8 | 45.6 | 35.8 | 55.1 |
| | *665k* | 1421.0 | 62.6 | 56.7 | 66.0 | 34.4 | 56.3 | 34.3 | 25.1 | 47.4 | 35.2 | 48.2 |
| | | ▲ 10.2 | ▼ 2.1 | 0.0 | ▲ 5.6 | ▼ 2.1 | ▼ 0.8 | ▲ 0.6 | ▲ 2.3 | ▼ 0.8 | ▲ 2.1 | ▼ 1.0 |
| VFL (Ours) | *150k* | 1357.1 | 60.8 | 55.8 | 66.5 | 36.9 | 53.9 | 30.6 | 28.9 | 45.4 | 36.9 | 52.6 |
| | *250k* | 1444.3 | 62.5 | 56.8 | 69.0 | 37.1 | 55.8 | 32.1 | 32.5 | 46.7 | 35.3 | 55.5 |
| | *350k* | 1462.6 | 63.7 | 58.0 | 69.5 | 37.1 | 57.0 | 33.4 | 33.2 | 47.0 | 36.6 | 50.8 |
| | *665k* | **1526.3** | **68.2** | **64.1** | **86.0** | **38.3** | **63.1** | **37.5** | **34.1** | **49.5** | **35.3** | 48.2 |
| | | ▲ 115.5 | ▲ 3.5 | ▲ 7.4 | ▲ 25.6 | ▲ 1.8 | ▲ 6.0 | ▲ 3.8 | ▲ 11.3 | ▲ 1.3 | ▲ 2.2 | ▼ 1.0 |

Table 2: **Comprehensive Benchmark Results for Data Ratio Experiments.** We compare data subset selection strategies—Oracle, Random, Expert [58], and our VFL—across sample sizes ranging from 150k to 665k. Results show that VFL consistently outperforms both Expert and Random baselines, with particularly notable gains on general, knowledge, and OCR tasks. In the table, results at the optimal 665k setting are **bolded**, while the best scores for other subset sizes are underlined.

novel approach that leverages our understanding of VFLs as a guiding principle for data curation. The core idea is that data instances most beneficial for improving MLLM capabilities are those that effectively engage, or are predicted to refine, these functionally specialized layers. By analyzing data "through the lens" of VFLs, VFL-Select aims to curate smaller, higher-quality, and more targeted datasets, prioritizing data based on its predicted utility.

**Experiment Setting.** We construct a diverse data pool consisting of 20 million vision instruction samples, covering a wide range of tasks and modalities. To implement VFL-Select, we first determine the functional value of a given sample $(\boldsymbol{x}, \boldsymbol{y})$ (input $\boldsymbol{x}$, ground-truth answer $\boldsymbol{y}$) in layer $k$ as:

$$R_k(\boldsymbol{x}, \boldsymbol{y}) = \frac{P(\boldsymbol{y} \mid \boldsymbol{U}(\boldsymbol{x})^{(\leq k)}, \boldsymbol{W})}{P(\boldsymbol{y} \mid \boldsymbol{U}(\boldsymbol{x})^{(\leq k-1)}, \boldsymbol{W})}, \tag{5}$$

where $P(\boldsymbol{y} \mid \boldsymbol{U}_{(\boldsymbol{x})}^{(\leq k)}, \boldsymbol{W})$ is the probability of generating $\boldsymbol{y}$ in Equ. 4. A higher value $R_k$ for a given sample suggests greater reliance on layer $k$ for correctly processing that sample, thus associating the sample with the vision function in that layer. This allows for a functional categorization of data without requiring explicit prior knowledge or semantic labeling of what kind of vision function each specific layer represents. In practice, we partition the entire dataset based on the highest $R_k$ score for each sample, effectively grouping data according to their dominant layer-wise influence. From each partition, we then uniformly sample data to construct balanced subsets for training. A crucial aspect for practical application is the scalability of this data classification process.

Notably, our findings indicate consistent VFL hierarchical trends across diverse MLLMs. *This allows the computationally intensive VFL-Select data classification to be efficiently executed using smaller proxy models* (e.g., TinyLLaVA-0.5B [75]), with the derived insights directly informing data curation for much larger target models (e.g., a 7B LLaVA model), substantially reducing computational overhead and enhancing VFL-Select's practical scalability.

**Experiment Results.** Tab. 2 demonstrates that VFL-Select consistently outperforms both Random and Human-Expert data selection strategies [58] across all tested subset sizes (150k to 665k instances). VFL-Select particularly excels on knowledge-intensive benchmarks. For instance, on SQA$^I$ with a 665k data subset, VFL-Select achieves a score of 86.0, substantially outperforming Random selection (e.g., 60.4) and Human-Expert selection (e.g., 72.1) as detailed in Tab. 2. This robust outperformance

| | Shallow-layer | | | | | | Deep-layer | | | | | |
|---|---|---|---|---|---|---|---|---|---|---|---|---|
| Method | $SQA^I$ | MMMU | $MME^P$ | POPE | SEED | Rel.(%) | VQAv2 | GQA | TextVQA | OCR | ChartQA | Rel.(%) |
| Full | 68.4 | 34.7 | 1476.9 | 86.4 | 67.3 | | 79.1 | 63.0 | 58.2 | 33.0 | 22.0 | |
| Random | 68.5 | 33.2 | 1483.0 | 84.7 | 62.2 | 97.3 | 75.7 | 58.9 | 55.3 | 30.3 | 19.7 | 93.1 |
| D2-Pruning [34] | 69.3 | 34.1 | 1391.2 | 85.7 | 63.1 | 97.4 | 73.0 | 58.4 | 51.8 | 30.9 | 20.3 | 92.0 |
| EL2N [41] | 65.5 | 34.0 | 1439.5 | 84.3 | 63.1 | 96.5 | 76.2 | 58.7 | 53.0 | 30.1 | 21.2 | 93.6 |
| COINCIDE [21] | 69.2 | 34.1 | 1495.6 | 86.1 | 63.8 | 99.0 | 76.5 | 59.8 | 55.6 | 29.1 | 20.8 | 94.0 |
| VFL (Ours) | 70.4 | 34.2 | 1504.2 | 86.1 | 63.5 | 99.5 | 77.4 | 61.4 | 57.1 | 31.0 | 22.0 | 97.4 |

Table 3: **Performance of Data Selection Methods using a 20% LLaVA-665k Subset.** All strategies, excluding "Full" (trained on 100% of LLaVA-665k), utilize only a 20% subset of the LLaVA-665k data for fine-tuning. Performance on shallow-layer and deep-layer task categories is presented relative to the "Full" model's scores (Rel.(%)).

confirms that VFL-Select efficiently identifies higher-utility data instances from large, heterogeneous pools, leading to enhanced model performance for a fixed data budget and demonstrating the value of leveraging VFL insights for intelligent data curation.

**Experiment on LLaVA-665k.** To further assess the versatility and effectiveness of our VFL-Select methodology, we conduct experiments focusing on its ability to identify high-utility data within a more constrained and established dataset, specifically LLaVA-665k [23]. The objective was to curate an optimal 20% subset from the LLaVA-665k dataset itself for fine-tuning. We compared VFL-Select against other selection strategies (Random, D2-Pruning, EL2N, COINCIDE) operating under this 20% data constraint. The performance of models fine-tuned on these subsets was evaluated relative to a model trained on the complete LLaVA-665k dataset ("Full"). As detailed in Tab. 3, with only 20% of the LLaVA-665k data, models fine-tuned using VFL-Select achieved 99.5% of the full-data performance on shallow-layer task benchmarks and 97.4% on deep-layer task benchmarks. These results significantly surpassed those of other data selection methods, underscoring VFL-Select's efficacy in identifying the most impactful training instances.

# 4 Related Work

**Layer-wise Representations in LLMs.** Recent studies investigate the role of individual layers within LLMs. AdaInfer [11] finds that many layers in LLMs are redundant, with only about 20% of layers being essential for general tasks and around 50% for sentiment analysis. Their method assesses the contribution of each layer by directly ablating it. Building on the same hypothesis, DSA [26] introduces a pruning strategy that leverages per-layer importance scores to search for a computation rule that determines the pruning ratio for each layer. In a similar vein, LISA [40] shows that many parameters introduced during LoRA fine-tuning are also redundant, and proposes selecting layers to fine-tune based on their weight norms.

**Layer-wise Representations in MLLMs.** There has been limited focus on layer-wise representations in MLLMs. The most relevant line of work comes from token pruning studies [5, 67, 72], which reveal that MLLMs do not require all vision tokens to perform accurate reasoning. Methods [67, 72] have shown that the acceptable token reduction rate varies across different layers. While prior work focuses on token-level efficiency, it leaves the reasons unexplained. In contrast, we investigate layer-wise vision functions, offering deeper insights into how vision tokens contribute across layer.

# 5 Conclusion

This paper reveals that Multimodal Large Language Models (MLLMs) develop narrow, hierarchical Vision Function Layers (VFLs) specialized for tasks like counting, localization, and OCR recognition. Using Visual Token Swapping and token dropping, we show these structures emerge consistently across models. Leveraging this, our VFL-targeted fine-tuning cuts parameter costs while preserving performance, and VFL-guided data selection achieves 98% of full-data results with just 20% of data. Our findings offer new paths toward more interpretable and efficient multimodal systems.

**Acknowledgement:** This work is supported in part by the National Natural Science Foundation of China under Grant No.62206174 and No.62576365, and Hong Kong Research Grants Council under NSFC/RGC Collaborative Research Scheme (Grant CRS_HKU703/24).

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

## A    Evaluation Protocol

In this section, we provide detailed evaluation settings. Our test set covers a wide range of capabilities, including general knowledge, visual recognition, spatial reasoning, as well as chart and our own tasks.

**Evaluation tool.**    We observed that the official evaluation tools provided by some benchmarks are inefficient in large-scale experiments. Therefore, we adopt two well-maintained toolkits: `lmms-eval` (v0.33) [3] and `VLMEvalKit` (v0.2) [4], to perform all our evaluations.

It is important to note that the version of the evaluation toolkit can significantly affect the final reported performance. For a fair comparison, especially for experiments that require external baselines (e.g., Tab.3), we follow the default evaluation protocol used by each benchmark. For all other experiments, we adopt `lmms-eval` and `VLMEvalKit` to improve efficiency. To ensure reproducibility, we release all the question prompts used during evaluation.

---

**Question Prompt**

**[ScienceQA$^I$]**
\nAnswer with the option's letter from the given choices directly.
**[MMMU]**
Multi-Choice = Answer with the option's letter from the given choices directly.
Open-Ended = Answer the question using a single word or phrase.
**[MME]**
\nAnswer the question using a single word or phrase.
**[POPE]**
\nAnswer the question using a single word or phrase.
**[SEED]**
\nAnswer with the option's letter from the given choices directly.
**[TextVQA]**
\nAnswer the question using a single word or phrase.
**[ChartQA]**
\nAnswer the question using a single word.

---

**CV-Bench.**    Unlike the default CV-Bench protocol, we report the final performance by directly averaging the results over the four subsets.

## B    More Results of on Vision Token Dropping Experiment

In this section, we present additional results from the vision token dropping experiments. We first provide a qualitative analysis on the *Math* dataset, followed by a comprehensive quantitative evaluation across multiple datasets by different MLLMs.

**Vision Function Layer in Different MLLMs.**    We provide additional experimental results on vision token dropping across various Multimodal Large Language Models (MLLMs) in Tab. 4. The models are categorized based on their vision encoders and language models as follows:

---

[3] https://github.com/EvolvingLMMs-Lab/lmms-eval
[4] https://github.com/open-compass/VLMEvalKit

| Vision Encoder & Language Model | General & Knowledge | | Recognition | Spatial | OCR |
|---|---|---|---|---|---|
| | SQA-I [33] | MMMU [68] | POPE [22] | CVBench [58] | ChartQA [35] |
| CLIP [43] & Vicuna [74]-7B | 68.8 | 34.7 | 86.4 | 56.2 | 22.0 |
| – drop 8 v-layers | 68.8 | 34.4 | 86.4 | 56.4 | 21.0▼ |
| – drop 16 v-layers | 68.7 | 34.3 | 86.0 | 53.7▼ | 15.9 |
| – drop 20 v-layers | 69.0 | 35.2 | 83.2▼ | 43.0 | 13.8 |
| – drop 24 v-layers | 65.7▼ | 33.9▼ | 38.1 | 37.2 | 12.8 |
| CLIP [43] & Vicuna [74]-13B | 72.7 | 35.4 | 85.9 | 53.0 | 22.6 |
| – drop 8 v-layers | 72.6 | 35.8 | 86.0 | 53.2 | 21.1▼ |
| – drop 16 v-layers | 72.7 | 35.4 | 84.9 | 53.2 | 15.8 |
| – drop 20 v-layers | 72.2 | 37.2 | 74.9▼ | 51.2▼ | 15.0 |
| – drop 24 v-layers | 70.1▼ | 34.6▼ | 11.5 | 52.2 | 13.5 |
| DINOv2 [39] & LLaMA-3 [14]-8B | 71.4 | 32.7 | 85.5 | 64.8 | 73.1 |
| – drop 8 v-layers | 70.1 | 34.0 | 85.0 | 66.4 | 70.8▼ |
| – drop 16 v-layers | 70.5 | 31.3 | 84.5 | 68.9 | 18.6 |
| – drop 20 v-layers | 69.6 | 34.8 | 78.2▼ | 66.5▼ | 13.4 |
| – drop 24 v-layers | 67.7▼ | 36.5 | 6.6 | 44.3 | 11.6 |
| DINOv2 [39] & Vicuna [74]-13B | 74.3 | 37.7 | 85.6 | 65.4 | 72.7 |
| – drop 8 v-layers | 74.3 | 40.7 | 84.6 | 66.0 | 62.0▼ |
| – drop 16 v-layers | 74.3 | 37.4 | 84.9 | 66.2 | 17.8 |
| – drop 20 v-layers | 74.6 | 37.2 | 76.9▼ | 57.3▼ | 14.0 |
| – drop 24 v-layers | 72.4▼ | 36.6▼ | 17.6 | 51.0 | 13.1 |
| Qwen-ViT & Qwen2.5-LM [3]-3B | 80.3 | 46.3 | 87.0 | 72.9 | 83.4 |
| – drop 4 v-layers | 80.0 | 46.9 | 86.9 | 72.9 | 78.6 |
| – drop 8 v-layers | 80.1 | 46.5 | 86.9 | 72.6 | 60.0 |
| – drop 16 v-layers | 79.6 | 46.2 | 82.7▼ | 56.4 | 13.0 |
| – drop 20 v-layers | 76.7▼ | 45.2▼ | 19.4 | 47.2 | 13.2 |
| Qwen-ViT & Qwen2.5-LM [3]-7B | 87.2 | 50.7 | 86.1 | 80.8 | 83.2 |
| – drop 4 v-layers | 87.4 | 50.8 | 86.3 | 81.0 | 82.7 |
| – drop 8 v-layers | 87.4 | 50.6 | 86.2 | 80.6 | 20.5 |
| – drop 12 v-layers | 87.2 | 50.2 | 79.5▼ | 69.1 | 17.4 |
| – drop 18 v-layers | 77.3▼ | 45.8 | 37.1 | 44.2 | 14.3 |

Table 4: **Vision Token Dropping on General Benchmarks.** $A_\downarrow$ indicates the onset of performance degradation, while $\underline{A}$ highlights significant drops. The results reveal a consistent hierarchical order of vision function layers across diverse MLLMs.

- **Vision encoders:** CLIP [43], DINO [39], and Qwen-ViT.
- **Language models:** Vicuna-7B/13B [74], LLaMA-3 [14] 8B, and Qwen2.5-LM-3B/7B [3].

Across different combinations of these components, all MLLMs consistently exhibit the phenomenon of a *Vision Function Layer*, showing similar trends. This demonstrates that the observed behavior is not an isolated case, but rather a shared internal mechanism among different MLLMs.

Interestingly, for MLLMs equipped with DINO as the vision encoder, we find that removing later-stage Vision Function Layers (VFLs) can actually enhance performance on certain CV-related tasks. On CV-Bench, for instance, dropping 20 layers results in higher accuracy than using the full model (68.9 vs. 64.8), indicating potential redundancy or task misalignment in deeper VFLs.

**Vision Token Dropping on MathVista.** We provide a qualitative analysis of vision token dropping on MathVista, illustrating how different vision token layers affect complex visual reasoning. For example, when interpreting a bar chart, the full model accurately read bar values (8 and 6, correctly finding none smaller than 6) and text. Dropping later vision token layers (VFLs-L20-L28) showed a nuanced impact. While the model still correctly reasoned that no bars were smaller than 6 (reporting values as 6 and 8), its OCR capability was severely compromised, outputting generic placeholders like "X-axis labels" instead of actual text. This suggests these layers are critical for fine-grained detail like text but less so for gross quantitative estimation. More aggressive dropping (VFLs-L10-L28) led to significant degradation in fundamental visual perception. The model then incorrectly identified bar values as approximately 4 and 2, wrongly concluding two bars were smaller than 6. These examples demonstrate that different layers uniquely contribute to diverse visual reasoning abilities essential for MathVista tasks, with later layers crucial for OCR and a broader range vital for accurate visual magnitude interpretation.

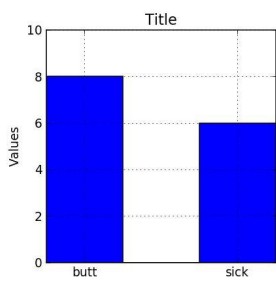

How many bars have values smaller than 6?

There are no bars in the chart that have values smaller than 6. The two bars shown in the chart have values of 8 and 6, respectively.

Drop VFLs-L20-L28

There are no bars in the chart that have values smaller than 6. The two bars shown in the chart have values of 6 and 8, respectively.

Drop VFLs-L10-L28

In the given bar chart, there are two bars that have values smaller than 6. The first bar has a value of approximately 4, and the second bar has a value of approximately 2.

What text can you read from the image?

The text in the image reads:- Title: Title- Values: 0, 2, 4, 6, 8, 10- Categories: butt, sick.

Drop VFLs-L20-L28

The text in the image reads:- "X-axis labels: X-axis labels"- "Y-axis label: Y-axis label"- "Title: Title.

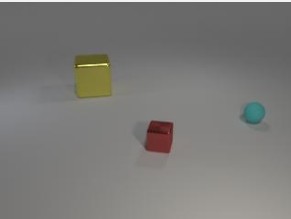

Subtract all large yellow metal blocks. Subtract all gray metallic cylinders. How many objects are left?

In the image, there are three objects:1. A large yellow metallic cube. 2. A small red metallic cube.3. A small cyan sphere. Since we need to subtract all large yellow metallic blocks and all gray metallic cylinders, we can see that there is one large yellow metallic block in the image. There are no gray metallic cylinders in the image. After removing the large yellow metallic block, the remaining objects are: 1. The small red metallic cube. 2. The small cyan sphere. So, there are 2 objects left.

Drop VFLs-L20-L28

In the image, there are two large yellow metal blocks and one small gray metallic cylinder. However, since the question asks to subtract all large yellow metal blocks and all gray metallic cylinders, we will remove these objects from the count. - There is 1 large yellow metal block. - There is 1 gray metallic cylinder. After removing these objects, the remaining objects in the image are: - 1 small gray metallic cylinder (which was not subtracted because it is not a large yellow metal block). - 1 small gray metallic cylinder (which was not subtracted because it is not a large yellow metal block). So, after subtracting the specified objects, there are no objects left in the image.

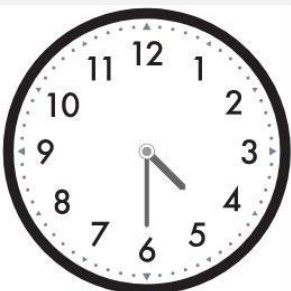

What time does the clock show?

> The clock shows 4:30. The hour hand is pointing at 4, and the minute hand is pointing at 6, which represents 30 minutes past the hour.

Drop VFLs-L25-L28

> The clock shows 4:30. The hour hand is pointing at 4, and the minute hand is pointing at 6, which represents 30 minutes past the hour.

Drop VFLs-L20-L28

> The clock shows 3:30.

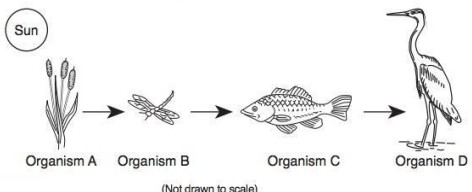

What would happen to organism c if organism b increased?

> In the food chain depicted in the image, Organism B (a dragonfly) is a predator of Organism A (a plant), and Organism C (a fish) is a predator of Organism B. If the population of Organism B increases, it means there will be more predators for Organism C. Here's what could happen: 1.... summary, if the population of Organism B increases, it is likely that the population of Organism C would decrease due to increased predation pressure and possibly resource competition.

Drop VFLs-L25-L28

> If the population of organism B (the herbivore) increases, it would likely have a positive impact on the population of organism C (the primary consumer). Here's why:1. **Increased Food Supply**: Organism B consumes organism C as its food source. If there is an increase in the number of organism B, there will be more consumers for organism C. This could lead to a higher demand for organism C, potentially increasing its population. 2. **Competition**: However, this increase in organism C might also lead to increased competition among the organisms for ....

Drop VFLs-L20-L28

> If the population of organism B increases, it could have several effects on organism C: 1. **Competition for Resources**: Organism B and organism C may compete for the same resources such as food, water, or space. An increase in the ...., an increase in the population of organism B would likely have a negative impact on organism C, potentially leading to a decrease in the population of organism C.

# C   Implementation Details of VFL-LoRA

In this section, we provide a detailed overview of our VLF-LoRA, including the training data, experimental setup, and implementation details.

**Training Data.**    We leverage SAT [46], a simulated spatial aptitude training dataset that includes both static and dynamic spatial reasoning tasks, comprising 175K question-answer (QA) pairs across 20K scenes. The dataset features three types of questions:

1. *Relative Spatial Relations*: determining whether object X is to the left, right, above, or below object Y, and identifying whether object A or B is closer to another object C.

2. *Relative Depth*: assessing whether object X is closer to the camera than object Y by computing object distances within the simulator.

3. *Counting*: determining the number of specific objects in a scene.

After filtering out multi-image questions, we retain a total of 151,724 training samples for use in our experiments.

**Experimental Setup and Implementation Details.**    We use two popular open-source MLLMs, Qwen2.5-VL-3B [64] and Qwen2.5-VL-7B [64], for our experiments. The experiments are conducted using a well-maintained GitHub repository[5]. We use 8 H20 GPUs, and the entire training process takes approximately 3 hours. For Qwen2.5-VL-7B, we apply LoRA fine-tuning to *layers 10 through 17, as well as layers 20, 21, 22, and 23*, while keeping all other parameters frozen. We set the LoRA rank to 32, the scaling factor (alpha) to 64, and the dropout rate to 0.05. For the 3B variant, we also conduct Vision Token Swapping experiments on the *Count* task. In this setting, LoRA fine-tuning is applied to *layers 16 through 26*, and all other experimental configurations are kept identical to those used for the 7B model.

**Detailed analysis on VFL-LoRA performance.**    Fig.3 in main paper shows that our VFL-LoRA method consistently outperforms standard LoRA and Reversed-VFL on both Qwen2.5-VL-3B and Qwen2.5-VL-7B models across In-Domain and Out-of-Domain tasks, while utilizing significantly fewer trainable parameters. For Qwen2.5-VL-3B, VFL (Ours) achieves an average In-Domain score of 83.5% (vs. 82.7% for LoRA) and an Out-of-Domain score of 72.9% (vs. 71.8% for LoRA), using only 0.9% parameters compared to LoRA's 3.1%. Notably, it improves the *Count* task to 72.3% and *MMMU* to 47.3%. On the Qwen2.5-VL-7B model, VFL (Ours) demonstrates even stronger advantages. It achieves an average In-Domain score of 85.0% (LoRA: 84.4%) and an Out-of-Domain score of 75.0% (LoRA: 74.3%). This superior performance is achieved with only 0.9% parameters, less than half of LoRA's 1.9%. Compared to Reversed-VFL, which uses a similar 0.9% parameters, our VFL method still shows clear improvements, particularly in the average In-Domain (85.0% vs 82.7%) and Out-of-Domain (75.0% vs 74.0%) scores. Key improvements include *Count* (72.6%) and *MMMU* (51.7%). These results underscore VFL-LoRA's effectiveness in enhancing model performance and generalization capabilities with remarkable parameter efficiency.

It is worth noting the performance on the *Distance* task, where VFL (Ours) (86.3% for 3B, 86.8% for 7B) does not consistently outperform the baselines (Reversed-VFL: 87.0% for 3B; LoRA: 88.3% for 7B). Our preliminary analysis suggests that the *Distance\** task might be predominantly language-driven. Tasks that rely more heavily on nuanced linguistic understanding or generation, rather than fine-grained visual feature manipulation targeted by VFL, may not benefit as significantly from our approach. In such language-centric scenarios, methods allowing broader adaptation of language components might yield more competitive results.

# D   Implementation Details of VFL-Select

In this section, we provide a detailed overview of our VLF-Select, including the data pool we collect, experimental setup, and implementation details.

---

[5] https://github.com/2U1/Qwen2-VL-Finetune

**Data Pool.** To simulate real-world data distributions, we begin with a large-scale data pool. Specifically, we construct this pool by merging all training data from Cambrian-1 [58], EAGLE-1 [51], and EAGLE-2 [51]. This diverse dataset provides broad coverage of scenarios encountered in practical applications.

**Experiment Setup and Implementation Details.** We use TinyLLaVA-1.5B-SigLIP [75] as the MLLM for data classification. Although its model size is smaller than the commonly used 7B models, it outperforms LLaVA-1.5-7B in our evaluations. Moreover, based on our observations, models of different sizes share the same ordering of vision function layers. This consistency allows the classification results obtained from the smaller model to be effectively transferred to the training of larger models. TinyLLaVA-1.5B-SigLIP consists of *22 layers*. We compute $R_k$ for $k = 8, 10, 12, 16, 20$ as defined in Equ.5. For each annotated sample, we group it based on the value of $k$ that yields the highest $R_k$, and then perform stratified sampling within each group according to $R_{22}$. This ensures that samples of varying difficulty levels are evenly represented in the selected subset. Note that although we compute $R_k$ six times, the computational cost remains low. This is because the dominant cost in MLLM inference comes from processing long sequences of image tokens. In our case, the image tokens are dropped after encoding, so computing multiple $R_k$ values incurs a cost comparable to a single pass with the full image tokens. We perform the data classification process on 16 H20 GPUs, which takes approximately 40 hours to complete. It is important to note that, as a generalizable preprocessing step, this feature extraction only needs to be done once and can be reused across different training settings. Therefore, we consider the computational cost to be acceptable. For the human-expert baseline, we follow the optimal dataset-type distribution reported in Cambrian [58] after multiple rounds of empirical tuning. Specifically, the composition is as follows: Language (21.00%), General (34.52%), OCR (27.22%), Counting (8.71%), Math (7.20%), Code (1%), and Science (1%).

**Detailed analysis on VFL-Select performance.** Tab.2 in main paper details a performance comparison between our VFL data selection strategy and a method based on human-expert curated data. This comparison spans various data subset sizes from 150k to 665k and covers a range of task categories. A clear and consistent trend indicates that VFL (Ours) generally achieves superior or highly competitive results when compared to Expert, with this advantage often becoming more pronounced as the volume of data increases.

Across general visual-language understanding tasks such as $MME^P$, $SEED^I$, and $GQA^I$, VFL (Ours) consistently surpasses Expert at all data quantities. For example, at the 665k data scale, VFL (Ours) scores 1526.3 on $MME^P$, 68.2 on $SEED^I$, and 64.1 on $GQA^I$, which are notably higher than Expert 's respective scores of 1421.0, 62.6, and 56.7. This pattern extends robustly into knowledge-intensive tasks ($SQA^I$, $MMMU^V$, AI2D) and OCR & Chart interpretation tasks (ChartQA, OCR, TextVQA). In the knowledge domain, VFL (Ours) demonstrates a particularly strong lead, with $SQA^I$ performance at 665k reaching 86.0 for VFL (Ours) versus 66.0 for Expert . Similarly, in OCR tasks, VFL (Ours) achieves 34.1 compared to Expert 's 25.1 at the 665k level, highlighting VFL's efficacy in selecting data pertinent to textual and schematic understanding. For instance, across these more reasoning-heavy categories, VFL (Ours) generally establishes a significant performance margin, suggesting its data selection is more effective for enhancing complex cognitive capabilities in models.

The comparison in vision-centric tasks, specifically "Count" and "Dist," reveals a more varied landscape. For the "Count" task, Expert shows a slight advantage at the smallest data size (150k: 38.3 vs. VFL's 36.9). However, with increasing data, VFL (Ours) tends to match or slightly exceed Expert , exemplified by scores of 35.3 (VFL) and 35.2 (Expert) at 665k. The "Dist" task shows fluctuating relative performance: VFL (Ours) leads at 150k (52.6 vs. 51.1) and 250k (55.5 vs. 46.8), while Expert is ahead at 350k (55.1 vs. VFL's 50.8). At the largest data point of 665k, both methods converge to an identical score of 48.2. It's noteworthy that the Oracle score for "Dist" (43.0) is unusually lower than many achieved scores, indicating potential specificities with this benchmark.

**ScienceQA$^I$ is considered an outlier in our dataset.** In our initial experiments, we retained at least the first 8 layers of vision tokens, based on the assumption that removing these early layers—which are responsible for fundamental visual recognition—would render the MLLM nearly non-functional. However, we found this assumption does not hold for ScienceQA$^I$. Specifically, the model achieves reasonable performance using only the first 4 layers of vision tokens.

As a result, in our data selection experiments, if no special handling is applied to the data required by ScienceQA$^I$, the final performance tends to plateau around 60. Remarkably, when we supplement the dataset with image-VQ samples that do not rely on vision tokens, the performance improves significantly to 86. This suggests that, for ScienceQA$^I$, vision tokens serve less as a source of rich visual information and more as a structural placeholder to maintain a unified input format for training.

## E   Discussion, Limitation and Future Work

**Limitations.**   *Task-Specific Benefits of VFL-LORA*: The advantages of VFL-LORA are most pronounced for tasks heavily reliant on the specific visual functions localized in the targeted layers. For tasks that depend more on language priors or statistical biases rather than detailed visual features (e.g., the CV-Distance sub-task), VFL-LORA offers limited gains compared to standard LoRA, which updates all layers. This suggests that the effectiveness of VFL-LORA is conditional on the nature of the downstream task and its alignment with the functions of the selected VFLs. *Scope of Investigated Visual Functions:* The current research primarily focuses on four key visual functions: OCR, Object Counting, Object Recognition, and Object Grounding. MLLMs are designed to handle a much wider array of visual concepts and reasoning types. The existence, localization, and hierarchical order of VFLs for other, potentially more abstract or complex, visual functions remain to be explored.

**Future Works.**   *Expanding the Repertoire of VFLs:* Future studies could extend the VFL analysis to a broader spectrum of visual questions beyond the four investigated, including more complex relational reasoning [73], open-world detection and recognition [48, 49], streaming-video understanding [70], 3D scene understanding [50] or even visual hallucinations problems [13, 42]. This would provide a more comprehensive map of functional specialization within MLLMs. *VFL-Guided MLLM Design:* The insights from VFL localization could directly inform the development of novel MLLM architectures. For instance, architectures could be designed with explicit layer specializations or routing mechanisms guided by VFL principles, potentially leading to models that are more interpretable, efficient, and easier to train for specific capabilities.

