# OpenReview forum: "Vision Function Layer in Multimodal LLMs"
_NeurIPS.cc/2025/Conference — NeurIPS 2025 poster_

### Official Review · Reviewer_5Arb · 2025-06-23

**Clarity:** 3
**Significance:** 3
**Originality:** 3
**Rating:** 5
**Confidence:** 5

**Summary:**

- The paper presents an analysis study about the importance of different LLM layers for various VQA tasks using MLLMs.
- The authors find that for different tasks, different layers are important and term those as the “Vision Function Layers” (VFL) for the Qwen-2.5-VL model.
- The authors also try finetuning a qen2.5-vl model with LoRA applied to only the VFLs and find it to perform at par or better than applying LoRA to all layers.
- Finally, they select samples with the highest value based on the probability based heuristic at the most important VFL for a given domain and show that using a subset of the dataset can result in fair performance

**Questions:**

- How does the findings change for video benchmarks? do we see the presence of VFL layers for video inputs too?
- How does VFL transfer to encoder-free MLLMs like chameleon or EVE-2/EVE?
- In the fig. 3 caption, the authors state: “contrasting with strong language priors for Distance and Relation.”, what is the basis of saying this? Relation and distance tasks do require good spatial reasoning abilities.

**Ethical Concerns:**

["NO or VERY MINOR ethics concerns only"]

**Final Justification:**

I believe the findings from this paper are helpful to the community and it may help in designing better MLLMs or even training a few layers for some tasks in future MLLMs.

**Limitations:**

The authors do not discuss any limitations but a few that come to mind:
- No analysis for encoder free MLLMs or video benchmarks.
- No analysis on different MLLM architectures (including MoEs).

**Quality:**

2

**Strengths And Weaknesses:**

### Strengths

- Good analysis which provides insights into what layers are important for different tasks.
- The VFL select method could be useful for future methods.

### Weaknesses

- There is little reference to previous works in the experiment section in order to support their findings. For example, previous works (like FastV and [vision-of-vlm](https://vision-of-vlm.github.io/)) have found that the attention scores to visual tokens is the highest in the earlier layers, so why do changing the visual tokens there does not affect the answer? This finding is counter-intuitive and should be explained.
- There is little effort to explain the findings of the paper about the presence of VFL layers, I would have liked the presence of some hypothesis.
- Analysis is only done with qwen2.5-vl model which is very strong model in itself. I would like to see how the findings change with different MLLMs like LLaVA-1.5, Cambrian-1 (multi-encoder approach), Intern2.5-VL and also an MoE based model like Deepseek-VL2. Because this is an analysis driven paper, multiple models should be analyzed to make the findings well grounded and widely applicable.
- How does VFL-select compare to methods using the probability of answers from the final LLM layer and also after the head? that is an important baseline which is missing. If the results are similar, VFL-select has lesser value with much analysis to be done for every different family of model.
- The finding about MMMU and ScienceQA-Img is well known (as found in MM-Star, I would have liked to see that citation and also evaluation on that benchmark).

---

> ### Author Rebuttal · Authors · 2025-07-30
>
> We thank the reviewer for the insightful comments and questions. Our response is as follows:
>
> > ***Q1 VFL still present on video benchmark:***
> >
> We further validated the presence of VFLs on Video-MME, which includes a variety of fine-grained tasks. Following the vision token dropping protocol described in Table 1 of the main paper, we applied this analysis to Video-MME. The results are presented below.
>
> | Video-MME | OCR Set | Spatial Perception | Counting | Object Recognition |
> | --- | --- | --- | --- | --- |
> | Full | 68.4 | 80.0 | 40.0 | 64.3 |
> | Drop-OCR-VFL | 38.6 | 83.3 | 38.4 | 64.5 |
> | Drop-Grounding/Spatial-VFL | 30.3 | 56.7 | 40.0 | 64.3 |
> | Drop-Count-VFL | 30.1 | 44.3 | 31.4 | 61.2 |
> | Drop-Recognition-VFL | 30.3 | 36.7 | 27.2 | 32.1 |
>
> As shown in the table above, **dropping the corresponding VFL leads to a sharp performance drop on the associated task set**, while performance on other task sets remains unaffected—consistent with the patterns observed in image-based benchmarks. For example, in the **OCR task** within Video-MME, performance drops significantly from **68.4 to 20.0** after removing the **OCR-VFL**.
>
> - **In summary, our results demonstrate that VFLs are not limited to static image inputs, but also extend to video-based tasks.** The persistence of VFLs in MLLMs when processing video inputs can be attributed to the fact that, fundamentally, a video can be viewed as a **long-context image**. Since extending the context length does not alter how MLLMs process visual tokens, the underlying behavior of VFLs remains unchanged. *(For more analysis on in-context and long-context settings, please refer to our response to Reviewer 5N5o.)*
>
> > ***Q2,W3,W2 VFL still present on Multi-encoder/Encoder-free/MoE-based MLLM:***
> >
>
> We further validated the presence of VFLs across a diverse set of MLLM architectures, demonstrating the generality of our findings. The results are presented below.
>
> | Multi-Encoder | MMMU | POPE | CVBench | ChartQA |
> | --- | --- | --- | --- | --- |
> | Cambrian-1 | 37.7 | 85.6 | 65.4 | 72.7 |
> | Drop-OCR-VFL | 40.7 | 84.9 | 66.2 | 17.8 |
> | Drop-Grounding/Spatial-VFL | 37.4 | 84.5 | 57.3 | 14.0 |
> | Drop-Recognition-VFL | 36.6 | 17.6 | 51.0 | 13.1 |
>
> | Encoder-free | MMMU | POPE | CVBench | ChartQA |
> | --- | --- | --- | --- | --- |
> | EVE-2 | 39.3 | 87.6 | 66.4 | 73.9 |
> | Drop-OCR-VFL | 39.3 | 85.6 | 66.2 | 20.5 |
> | Drop-Grounding/Spatial-VFL | 38.2 | 86.4 | 43.0 | 17.3 |
> | Drop-Recognition-VFL | 38.2 | 11.5 | 37.2 | 14.3 |
>
> | MoE-Based | MMMU | POPE | CVBench | ChartQA |
> | --- | --- | --- | --- | --- |
> | Deepseek-VL2 | 40.7 | 88.7 | 69.5 | 86.0 |
> | Drop-OCR-VFL | 40.7 | 88.7 | 69.2 | 40.4 |
> | Drop-Grounding/Spatial-VFL | 39.5 | 88.4 | 33.5 | 23.5 |
> | Drop-Recognition-VFL | 40.7 | 23.5 | 33.5 | 22.4 |
>
> As shown in the table above, we examined representative models with different architectures and conducted experiments on datasets from four task categories. The results demonstrate that **VFLs are consistently present across MLLMs with diverse architectural designs**.
>
> - **In summary, our findings regarding VFLs are robust and generalizable across model families.** We chose **Qwen2.5-VL** as the primary model for our main experiments because it is one of the most widely used, high-performing, and representative open-source MLLMs available. Additionally, in the Appendix, we provide experimental results on other model families (e.g., **LLaVA**), and the **VFL findings remain consistent** across these different MLLM architectures.
> - **We hypothesize that the emergence of VFLs** in MLLMs reflects an internal hierarchical structure of visual processing. This is analogous to the human visual system, where perception proceeds through layered stages—from low-level feature extraction to high-level semantic reasoning—with each layer supporting the next. Similarly, MLLMs may allocate specific visual functions to distinct layers, such that a given capability (e.g., OCR or object counting) only emerges after prerequisite features are computed. This view is also consistent with findings in LLMs [1,2], where textual information is processed in a **coarse-to-fine**manner across layers. **We believe this hypothesis provides a principled explanation** for the layer-specific specialization observed in VFLs.
>
> > ***Q3,W5 Language priors in CVBench and Results in MM-Star***
> >
>
> - The statement **"strong language priors for Distance and Relation"** in **Figure 3** refers to the **Distance** and **Relation** subsets of CV-Bench. For these two tasks, the model achieves significantly higher-than-random accuracy even **without access to visual tokens**. For example, on the **Distance** task, the MLLM exceeds random guess performance by over **10 percentage points** without any visual input. This indicates that the model leverages strong **language priors**, meaning it can often answer these questions correctly **without actually interpreting the image**.
> - We further validated the presence of VFLs in MM-Star
>
> |  | MM-Star | SEED | ScienceQA | MMMU |
> | --- | --- | --- | --- | --- |
> | Full | 63.9 | 77.6 | 87.2 | 50.7 |
> | Drop-OCR-VFL | 64.0 | 77.5 | 87.4 | 50.6 |
> | Drop-Recognition-VFL | 57.6 | 52.4 | 77.3 | 45.8 |
>
> Since MM-Star is a composite benchmark constructed from multiple datasets (e.g., SEED, ScienceQA, MMMU, etc.) with samples filtered to exclude those that can be answered without visual information, we also report performance on the original datasets for comparison. As shown, the VFL phenomenon still holds on MM-Star. For example, since the constituent datasets of MM-Star do not require OCR capabilities, dropping the OCR-VFL has no impact on performance, consistent with our expectations.
>
> > ***W4 VFL-Select Baseline***
> >
>
> Using only the final-layer output for selection performs on par with random choice, showing no significant advantage. We observed that relying solely on the final layer introduces dataset selection bias, causing performance to fall below that of random selection. This observation is corroborated by the results in [3].
>
> > ***W1  Clarity on the VFL***
> >
>
> - **Our findings are consistent with prior studies on attention in early layers.** Thank you for the insightful comment. Our findings do *not* contradict prior works such as **FastV**. As these works [1,2] have noted, early layers primarily perform **feature extraction and fusion**, which explains the high attention scores to visual tokens in shallow layers. This is a natural part of the perception pipeline and does not conflict with our results.
> - **Our analysis focuses on vision function rather than attention patterns.** While early layers may attend strongly to visual tokens, attention magnitude does not imply that a layer is functionally responsible for the final prediction. Our **VFL framework** is based on **causal intervention**—we perturb visual tokens at a specific layer while keeping others fixed. If swapping or dropping tokens at that layer does not change the output, it indicates that the required visual function is not processed there. The fact that modifying early-layer tokens does not affect the answer supports this view: these layers handle general encoding, while task-specific visual reasoning emerges in deeper layers. We will clarify this distinction and cite related works in the revised version.
>
> [1] Not All Layers of LLMs Are Necessary During Inference. 2024
>
> [2] Zero-Shot Vision Encoder Grafting via LLM Surrogates ICCV2025
>
> [3] Concept-skill transferability-based data selection for large vision-language models.

---

> > ### Comment · Reviewer_5Arb · 2025-08-03
> >
> > Thank you for the rebuttal! I have one remaining question for the authors:
> >
> > With the presence of VFL confirmed for video benchmarks too, how can the findings from this paper be used to develop/train better future MLLMs? For example, would they recommend only training VFL layers for the associated task samples in a dataset or go some other way?

---

> ### Author Response · Authors · 2025-08-03
> **Response to Reviewer 5Arb**
>
> Thank you for your quick response! We are pleased to further discuss on an important and promising direction: **leveraging the understanding of Vision Function Layers (VFLs) to advance MLLM design and training.**
>
> - **Development for Better MLLMs.** VFLs enables a new perspective on **test-time scalability** beyond conventional token-level. While recent work focuses on scaling computation by generating more reasoning tokens—each of which passes through a fixed model depth—our findings introduce a novel **function-level scaling paradigm**. The model can **selectively re-invoke specific vision function layers** that are relevant to the task at hand. For example, in our experiments on OCRBench, running the OCR function layer (layer 22 in Qwen2.5-VL) **twice**—once as normal and once again after the full forward pass—boosted performance from **82.2 to 86.1**, effectively increasing model capacity without modifying the architecture. This demonstrates a new, interpretable, and adaptive computation strategy that unlocks a **new axis for scaling model performance** beyond token count.
> - **Training for Better MLLMs.** As you suggested, VFLs can guide a more efficient training paradigm. Using **VFL-Select**, we categorize data by required visual skills, and fine-tune only the corresponding **VFL-layers** via **VFL-LoRA**. Current MLLM training, especially vision instruction tuning, is compute-intensive. Our approach reduces overhead by avoiding full-model fine-tuning while improving performance (see **Fig. 3**). It also enables more interpretable and modular training tailored to each vision function.
>
> The directions discussed above are all **validated and feasible pathways** for improving current MLLMs, based on our earlier empirical findings. Beyond these, we believe there exist many **exciting and unexplored opportunities** that can be pursued in future research. We look forward to further discussions and greatly appreciate your continued insights.

---

> > ### Comment · Reviewer_5Arb · 2025-08-04
> >
> > Thank you! It sounds exciting and I look forward to the community's work. I will raise my score.

---

> > > ### Author Response · Authors · 2025-08-04
> > > **Response to Reviewer 5Arb**
> > >
> > > We sincerely thank the reviewer for the encouraging recognition. Your feedback is truly motivating. We will continue to explore the potential of VFLs and stay engaged with the community’s evolving needs.

---

### Official Review · Reviewer_TCaw · 2025-06-29

**Clarity:** 3
**Significance:** 2
**Originality:** 3
**Rating:** 4
**Confidence:** 4

**Summary:**

The paper investigates vision-related functional decoding in Multimodal Large Language Models (MLLMs), revealing that visual functions like counting, grounding, and OCR are localized within narrow decoder layer blocks, termed Vision Function Layers (VFLs).

Using a novel Visual Token Swapping framework, the study finds consistent VFL depth and order across MLLMs, aligning with human visual processing (recognition → counting → grounding → OCR).

The insights enable practical applications: VFL-LoRA, a targeted fine-tuning method, outperforms full LoRA by focusing on relevant VFLs.

**Questions:**

1. Experiments focus on basic visual functions (such as recognition and counting), whereas there are numerous complex and diverse tasks in the real world. The VFL theory has not been validated in these real-world tasks.
2. This paper proposes the visual token swapping and token dropping strategy to validate the VLF theory. However, many papers [1] drop vision tokens directly, according to the attention map or other strategies. Since these visual tokens can be directly discarded, can your operation still prove your theory?
3. Please provide a more detailed explanation of how  visual token swapping and token dropping are specifically implemented.


[1] An Image is Worth 1/2 Tokens After Layer 2: Plug-and-PLay Acceleration for VLLM Inference

**Ethical Concerns:**

["NO or VERY MINOR ethics concerns only"]

**Final Justification:**

The author's response effectively addressed my concerns. Based on the issues from other reviewers and rebuttal from author, I have decided to improve my rating.

**Limitations:**

Yes

**Quality:**

2

**Strengths And Weaknesses:**

# Strengths
1. The Visual Token Swapping and Dropping methods provide a systematic way to probe layer-specific visual functions in MLLMs, enabling precise localization of functions like OCR, counting, and recognition to narrow layer blocks. This framework addresses the "black box" problem of MLLMs and sets a new standard for interpretability research.
2. VFL-LoRA reduces tunable parameters by ~50% while improving out-of-domain generalization compared to standard LoRA
3. The study validates findings across 8+ benchmarks (e.g., ScienceQA, MMMU) and uses controlled image pairs to isolate single visual functions, ensuring robust results.

---

> ### Author Rebuttal · Authors · 2025-07-30
>
> We thank the reviewer for the insightful comments and questions. Our response is as follows:
>
> > ***Q1. VFL has been successfully validated on real-world tasks, demonstrating its practical effectiveness.***
> >
>
> - **We systematically validated the presence of Vision Function Layers across diverse real-world tasks.**
>     - In Table 1 of the main text, we validate the presence of Vision Function Layers across nine widely used MLLM benchmarks—spanning General & Knowledge, Recognition, Spatial, OCR, and Chart categories—thereby encompassing a comprehensive range of vision functions. This broad empirical coverage has also been recognized by other reviewers, who noted that the results are “exhaustive with a wide coverage of task types and datasets.”
>     - Our pipeline is extensible to a wide range of vision functions, as discussed in our response to Reviewer Sn1h, W2, Q1 ("More visual functions"), particularly in the paragraph of W2, where we demonstrate how to systematically identify additional Vision Function Layers using our proposed framework.
> - To further demonstrate the applicability of our approach to real-world tasks, we evaluate our method under the following scenarios:
>     - **Fundamental vision capabilities serve as the essential foundation for tackling complex visual reasoning tasks.**  We have identified six distinct Vision Functions—namely OCR, Object Relation, Grounding, Count, Emotion Recognition, and Recognition—that collectively encompass a wide spectrum of common visual capabilities. The compositional integration of these atomic vision functions facilitates the resolution of more complex and higher-order tasks.
>     - **VFLs remain valid under different prompting techniques**, including in-context learning, chain-of-thought (CoT) prompting, and long-context settings. For details, please refer to our responses to Reviewer 5N5o: W2 *“VFL Still holds on in-context learning and long-context”* and W3 *“Application in CoT and GRPO.”*
>     - **VFLs also hold when the input modality is video rather than static images.** For supporting evidence, please refer to our response to Reviewer 5Arb, Q1 *“VFL still present on video benchmark.”*
>
>
> > ***Q2. The concept of Vision Function Layers (VFL) offers a theoretical justification for inference acceleration techniques that leverage selective vision token dropping.***
> >
>
> - **Visual tokens can NOT be directly discarded.** For token reduction methods like FastV, at least 25% of the tokens must be retained to maintain final performance; discarding all tokens directly results in significant performance degradation.
> - **VFL addresses fundamentally different concerns compared to token reduction techniques.** VFL focuses on the **vertical** behavior of vision tokens across each Transformer layer in MLLMs, analyzing the functional role of each layer by dropping ALL tokens at specific layers. In contrast, token reduction methods address the **horizontal** redundancy among vision tokens within a single layer, exploiting this redundancy to selectively drop tokens and accelerate inference.
> - **VFL offers a theoretical justification for *acceleration methods based on* token reduction.** For token reduction methods such as FastV, achieving optimal pruning efficiency requires carefully selecting parameters such as the starting Transformer layer for pruning and the proportion of tokens to drop. These hyperparameters are typically determined through empirical tuning and are not easily adaptable to different datasets or task types. In contrast, our discovery of Vision Function Layers (VFLs) provides a principled framework to guide such decisions. For example, as shown in Table 1 of our paper, for knowledge-based tasks, all vision tokens can be safely discarded starting from layer 10, offering clear guidance for layer-wise pruning strategies.
>
> > ***Q3.* Detailed explanation of Vision Token Swapping and Vision Token Dropping**
> >
>
> Both Vision Token Swapping and Vision Token Dropping are implemented at the decoding stage of the MLLM's generation process. Specifically, during inference, we operate directly on the **key-value (KV) cache** corresponding to the image tokens.
>
> - **Vision Token Swapping:** For a given layer, we replace the KV cache entries corresponding to the image tokens of the original input with those from a paired image. This intervention effectively simulates how the model would behave if it were processing a different visual input at that specific layer, while keeping all other layers and tokens unchanged.
> - **Vision Token Dropping:** Similarly, we remove the KV cache entries corresponding to the vision tokens at a given layer, effectively nullifying the visual input at that layer while leaving the rest of the computation unchanged. This allows us to isolate the contribution of visual information at different layers.
>
> These interventions are **causal** and **non-invasive** with respect to the rest of the model pipeline, and enable precise probing of where and how visual functions are executed across layers.

---

> ### Author Response · Authors · 2025-08-04
> **Official Comment by Authors — Welcoming Further Discussion with Reviewer TCaw**
>
> Thank you for acknowledging our response. We hope **our clarifications—the fundamental differences** between our VFLs and token pruning methods, along with our framework details and the **additional experiments (covering more vision functions, in-context learning, complex CoT prompting, and video benchmarks)**—have sufficiently addressed your questions. If so, we would greatly appreciate your reconsideration in the final evaluation.
>
> Should you have any further questions or comments during the discussion period, we’ll be happy to respond promptly. Thank you again!

---

> > ### Author Response · Authors · 2025-08-06
> > **Respectfully Requesting Final Comments Prior to Discussion Conclusion**
> >
> > Dear Reviewer TCaw,
> >
> > As the discussion period progresses, we would greatly appreciate any further comments or suggestions you may have. We hope our responses have addressed your concerns and would be glad to clarify anything further if needed.
> >
> > Thank you very much for your time and consideration.
> >
> > Sincerely,
> >
> > The Authors

---

### Official Review · Reviewer_Sn1h · 2025-07-02

**Clarity:** 2
**Significance:** 2
**Originality:** 3
**Rating:** 4
**Confidence:** 4

**Summary:**

This paper investigates the internal "black box" mechanisms of visual processing in MLLMs. The authors propose an analytical framework called Visual Token Swapping, which modifies targeted KV cache entries to precisely identify the function of specific layers during decoding. A core discovery is the existence of Vision Function Layers, which are narrow blocks of 2-3 layers where specific visual functions like counting or OCR are exclusively handled. These VFLs exhibit a consistent hierarchical order across different MLLMs. Leveraging these insights, the paper introduces practical applications for model optimization. VFL-LORA applies LoRA only to relevant VFLs, outperforming full LoRA in out-of-domain generalization while using fewer parameters. Furthermore, VFL-select, an automated data curation strategy, surpasses human experts and achieves 98% of full-data performance with only 20% of the original dataset. This work provides a deeper understanding of MLLM visual processing, enabling more efficient and interpretable models.

**Questions:**

1. The authors could incorporate a wider range of visual functions into the framework, as MLLMs exhibit complex visual capabilities and therefore require a more comprehensive analysis.

2. The authors should design corresponding benchmarks to evaluate different visual functions, since current MLLM benchmarks usually rely on multiple visual capabilities.

**Ethical Concerns:**

["NO or VERY MINOR ethics concerns only"]

**Final Justification:**

The authors' rebuttal has largely addressed my concerns. Consequently, I will adjust the rating to borderline accept.

**Limitations:**

yes

**Quality:**

2

**Strengths And Weaknesses:**

Strengths:
1. The paper is clear, structured, and effectively explains the experimental design and results. The writing is concise yet comprehensive.
2. This paper provides a novel perspective on understanding MLLMs by introducing the concept of "Vision Function Layers". This perspective offers a novel approach to explain how MLLMs process visual inputs and presents new insights into how these models function internally.

Weaknesses:
1. Lack of technical contributions. While this paper introduces an interesting framework, it does not present sufficient technical contributions. The primary contribution lies in the analysis of vision functions across layers, rather than offering technical innovations that could push the field forward. This limits the technical impact of the paper.
2. Limited scope of visual functions. The paper only considers four visual functions (recognition, counting, grounding, and OCR), which is too narrow. There are many other important visual reasoning tasks, such as object relationship inference and emotion recognition, that are not covered. This limited scope means the analysis may not fully capture the complexity of visual functions in MLLMs.
3. The analysis is based solely on the contribution of individual layers, without considering the interaction between multiple layers. In reality, the functionality of different layers may interact and jointly influence the model's performance. By ignoring this interaction, the study may not provide a complete understanding of how visual features are processed and integrated across layers.
4. The evaluation benchmarks used in the paper do not rely solely on single visual functions, which may undermine the accuracy of the results in assessing specific vision functions.

---

> ### Author Rebuttal · Authors · 2025-07-29
>
> We appreciate the reviewer’s comments. For better clarity, we respond in a reorganized order rather than the original question sequence.
>
> > ***W2, Q1. More visual functions.***
> >
>
> ***More visual functions can be easily identified using our pipeline.** W*e truly appreciate your positive assessment of our work on VFLs and the framework we proposed. One of the key contributions of our paper is **the discovery and localization framework for VFLs**. With the help of this framework, one can easily identify and analyze any vision function of interest, detailed as follows:
>
> - **Two additional VFL.** As an additional demonstration, we present two more VFLs, namely the **Object Relation Layer** and the **Emotion Recognition Layer**, as suggested by the reviewer. We simply follow the procedure outlined in **Section 2.3** and replace the paired images accordingly to identify any vision function layer (VFL) of interest. Specifically:
>     - **(1) Object Relation Pairs:** Adapted from the CV-Bench Depth Order benchmark [1], this task requires the model to determine which of two distinct objects is closer to the camera. The **change rate** is defined as the proportion of prediction changes after swapping vision tokens with those from an empty image. For **Qwen2.5-VL-7B**, the change rate peaks at **layer 20**, indicating that object relation reasoning is primarily handled at this specific layer—consistent with the patterns observed in other VFLs.
>     - **(2) Emotion Recognition Pairs:** We first use **Qwen3** [2] to generate 200 scene descriptions involving different emotions, including *joy, sadness, anger, fear,* and *disgust*. Based on these descriptions, we use **FLUX.1** [3] to generate corresponding images. We then randomly pair images representing different emotions and formulate multiple-choice questions to query the model. The **change rate** is calculated as the proportion of prediction changes after swapping vision tokens between images depicting different emotions. For **Qwen2.5-VL-7B**, the change rate peaks at **layer 10**, indicating that emotion recognition is primarily handled at this specific layer—consistent with the patterns observed in other VFLs.
>
>     |  | Peak Layer | Peak Value | Other Layer Average |
>     | --- | --- | --- | --- |
>     | OCR-VFL | 22 | 92.8% | 0% |
>     | Object-Relation-VFL | 20 | 81.5% | 7% |
>     | Grounding-VFL | 18 | 100% | <5% |
>     | Count-VFL | 14 | 87.4% | <5% |
>     | Emotion Recognition-VFL | 10 | 91% | <5% |
>     | Recognition-VFL | 0-8 | 90% | 37% |
> - **Any vision function of interest.** It is important to note that all models used in the above process are publicly available and open-sourced. By leveraging an LLM to generate prompts and using an image generation model to create paired images, **this pipeline—as described in the main paper—provides a general and reproducible approach to localizing any vision function of interest.**
> - **More analysis for different VLFs.** We summarize the layer-wise localization of the six identified VFLs on **Qwen2.5-VL-28-layer**, along with their **change rate distributions** across all layers. The results (table above) show that for each vision function, the corresponding **VFL** exhibits a distinct peak in change rate, while the function remains largely inactive in other layers—indicating a clear specialization of layers for different visual capabilities. As indicated in the supplementary material, we consistently observed this phenomenon across all tested MLLMs. We hope that the additional VFLs we provided, along with the proposed pipeline for broadly discovering VFLs, have effectively addressed your question.
>
> > ***W4, Q2. Each experiment is designed to ablate a single visual capability:***
> >
>
> In this work, we propose two approaches for analyzing and localizing VFLs: **vision token swapping** on our custom-designed benchmarks, and **vision token dropping** on carefully-selected MLLM benchmarks. They are all designed to ablate a single visual capability.
>
> - ***Vision token swapping ablates a single visual capability using paired images.** W*e construct **paired images that differ only in one specific visual function** being tested. When vision features are swapped at a particular layer and the model's prediction changes to match that of the swapped image, the **change rate** indicates that the corresponding visual capability is executed at that layer. Since the paired images are carefully designed to differ solely in the targeted visual function, this setup effectively rules out interference from multiple visual capabilities. For example, in the **OCR pairs**, we vary only the text content, while keeping the font, font size, font color, and text position identical. This ensures that only the OCR capability is being evaluated, without interference from other visual factors.
> - **Vision token dropping ablates one core visual capability through carefully selected dataset types, validated by experimental results.**
>     - The commonly used MLLM benchmarks (Table 1) we selected are each primarily designed to evaluate one core **visual capability**. This categorization is also supported by [1], which groups these datasets into four categories: *General & Knowledge*, *Recognition*, *Spatial*, and *OCR & Chart*.
>     - **The sharp performance drop (rather than a gradual decline) in Table 1 further validates the necessity of only one core visual function required by these benchmarks.**  Specifically, the accuracy on these benchmarks does **not gradually decline** as vision tokens are dropped; instead, a **sharp performance drop** is observed precisely when the vision tokens from the corresponding **VFL** are ablated. *(As a comparison, see our response to Reviewer 5N5o regarding A-OKVQA, where performance gradually declines as multiple VFLs are dropped, due to the task requiring a combination of vision functions.) For example, OCR-related benchmarks show a sharp drop in performance after removing the OCR layer (layer 22 in Qwen2.5-VL-7B), while others remain unaffected. Similarly, each benchmark only degrades after its corresponding VFL is dropped. In contrast, general knowledge tasks rely less on visual input and perform well as long as basic recognition VFLs are intact. These results provide strong and comprehensive support for our formulation of Vision Function Layers (VFLs).*
>
> > ***W3. Interaction between different VFLs.***
> >
>
> - **The effects of different VFLs are largely independent of one another.** Thank you for raising this suggestion—this is indeed a very interesting direction. In our preliminary exploration, we observed that the activation of different VFLs tends to follow a **linear additive pattern**, with each VFL operating **relatively independently**. As discussed in the main paper, specific vision functions are localized to **dedicated layers**, further highlighting the importance of understanding each **atomic VFL** individually.
> - **Atomic VFLs can be interacted and composed to handle complex tasks.** We have also explored the application of different VFLs in Chain-of-Thought (CoT) and GRPO settings in our response to **Reviewer 5N5o**—please refer to that response for more details. While we have observed instances where multiple VFLs are engaged within a single task, developing **systematic methods to explore their interactions**—such as building complex tasks that require coordinated visual functions and solving them step-by-step—remains a promising direction for future work. In this paper, our focus is to **discover and define** the concept of VFLs—a previously overlooked but fundamental phenomenon in MLLMs.
>
> > ***W1. Technical Contributions:***
> >
>
> - **To the best of our knowledge, we are the first to discover, define, and provide a systematic toolset for understanding Vision Function Layers (VFLs)** in MLLMs. This work establishes VFLs as a fundamental structural and functional unit in vision-language processing—an aspect previously overlooked in the literature.
> - **Our contributions span from VFL discovery framework and foundational insights to practical applications.** Building upon this foundation, we introduce two practical extensions: **VFL-LoRA**, a parameter-efficient fine-tuning method targeting specific VFLs, and **VFL-Select**, a technique for automatic dataset construction based on visual function needs. **It even outperforms methods specifically designed for these particular tasks!**
> - More broadly, our work provides an end-to-end perspective that spans the entire MLLM development process—from data selection and model training to the interpretation of internal mechanisms. Concretely, we address key questions: how to select training data, how to train models based on targeted visual functions, and how to understand the model’s internal behaviors. This comprehensive understanding not only clarifies how current MLLMs function but also offers practical insights for developing the next generation of more effective and interpretable multimodal models.
>
> We believe these contributions are technically substantial, novel, and impactful. We hope the reviewer finds them meaningful.
>
> [1] Cambrian-1: A Fully Open, Vision-Centric Exploration of Multimodal LLMs.
>
> [2] Qwen3 Technical Report.
>
> [3] FLUX.1 Kontext: Flow Matching for In-Context Image Generation and Editing in Latent Space.

---

> ### Author Response · Authors · 2025-08-05
> **Official Comment by Authors**
>
> Dear Reviewer Sn1h,
>
> As the discussion period is approaching its end, we welcome any further comments and suggestions on our work from you to see if our responses solve your concerns. Thank you once again for your time and valuable feedback!
>
> Best regards,
>
> The Authors

---

> > ### Comment · Reviewer_Sn1h · 2025-08-06
> >
> > The authors' rebuttal has largely addressed my concerns. Consequently, I will adjust the rating to borderline accept.

---

> > > ### Author Response · Authors · 2025-08-06
> > > **Official Comment by Authors**
> > >
> > > Thank you for your positive feedback and for your engagement throughout the review process! We are glad to hear that your concerns have been addressed. The details discussed in our response will be carefully reflected in the revised version of the paper. If you have any further questions or need additional clarification, please feel free to let us know.

---

### Official Review · Reviewer_5N5o · 2025-07-02

**Clarity:** 4
**Significance:** 3
**Originality:** 3
**Rating:** 4
**Confidence:** 4

**Summary:**

The paper provides a nice experiment design to assess where reasoning takes place in VLMs. The token dropping experiment is simple yet effective in isolating and pinpointing the layers with the most influence on a vision subtask. Empirically, the authors test their hypothesis on 4 different types of tasks - OCR, Grounding, Counting, and Recognition, and present insights about the particular layer where the task is "performed". Next, they show how fine-tuning only the pinpointed layers improves performance, reaffirming the insight. Finally, the authors utilize this insight into data selection for pre-training VLMs and achieve almost similar performance with ~20% of the data.

**Questions:**

Refer Weakness.

**Ethical Concerns:**

["NO or VERY MINOR ethics concerns only"]

**Final Justification:**

I have read the rebuttals, and I maintain my ratings.

**Limitations:**

Yes.

**Paper Formatting Concerns:**

None.

**Quality:**

3

**Strengths And Weaknesses:**

Strengths:

1. Interpreting the inner workings of VLMs is an important topic with significant interest recently.
2. The paper is very clear with a simple but intuitive methodology.
3. Empirical results are exhaustive with a wide coverage of task types and datasets.
4. Pre-training data selection is a smart application.

Weakness:

1. Only low-level tasks considered: Currently, the tasks are simple enough to be effective through direct prompting strategies. A lot of the insights presented are novel but do not take into account the high-dimensional vision-language interplay, which elicits the majority of VLM reasoning. For example, for each task, the prompt itself is not the most important aspect of the representations - OCR, recognition, etc., can be performed directly through probing vision tokens. It would be significantly impressive if these insights hold onto more difficult tasks like VQA (like on datasets like A-OKVQA), where VLMs truly shine.

2. Building on this, I would be excited to see how prompting techniques change the visual functions across layers. For example, consider in-context learning, where $k$ exemplars are pre-appended to the prompt. Now there are $k$ times more visual tokens - how does the performance change then? How about prompts with longer context?

3. The authors should also evaluate their approach on step-by-step prompting techniques like Chain-of-thought (CoT) reasoning, where a big visual reasoning task is broken down into simpler steps. If the author's findings confirm that each step of CoT activates similar layers, it will be a very deep contribution as multiple future works like PPO/GRPO can benefit from layer-specific policy learning.

Typos:

Section 2.1 "Multi-Model Large Language Models"

---

> ### Author Rebuttal · Authors · 2025-07-28
>
> We thank the reviewer for the insightful comments and questions. Our response is as follows:
>
> > ***W1. VFL Still holds on high-dimensional vision-language task***
> >
>
> Following your suggestion, we conducted both qualitative and quantitative vision token dropping experiments on A-OKVQA, a dataset that includes questions requiring reasoning over various types of knowledge such as commonsense, world knowledge, and visual understanding.
>
> | Qwen-2.5-VL-7B | Full | Drop OCR-VFL (18-28 layers) | Drop OCR-Gounding-Count-VFL (12-28 layers) | Drop (6-28 layers) |
> | --- | --- | --- | --- | --- |
> | A-OKVQA | 89.25 | 88.94 | 76.68 | 57.90 |
>
> - **Foundational VFLs remain critical even in complex tasks that need high-dimensional vision-language interplay.**  As shown above, removing **OCR-VFL** has no impact on performance, as A-OKVQA does not include OCR-related questions. In contrast, dropping **Grounding-Count-VFL** causes a 10% drop, with errors concentrated in questions like *“How many people will dine at this table?”*, which require counting objects (e.g., bowls) before reasoning. Even after partially removing **Recognition-VFL**, the model correctly answers questions like *“What does the man who sits have trouble doing?”* (*“Walking”*), relying mainly on language priors. In summary, **VFLs remain active and functionally aligned** even in complex tasks requiring multi-step vision-language reasoning.
> - **Atomic VFLs can integrate with language priors to support reasoning.**  These results are also intuitively reasonable, as **VLMs tend to process visual and language cues relatively independently**. For example, we newly introduced two VFLs—Emotion Recognition and Object Relation (see detailed information in our response to Reviewer Sn1h)—and found that the visual version of the *Sentiment* task, which has been shown in [1] to be handled in the middle and deeper layers of LLMs, is instead processed in the early layers of VLMs. This discrepancy not only highlights the fundamental differences between LLMs and VLMs, but also underscores the significance of identifying and analyzing **Vision Functional Layers**.
>
> > ***W2. VFL Still holds on in-context learning and long-context:***
> >
>
> Following your suggestion, we used VL-ICL Bench [2] to evaluate in-context learning. Specifically, we tested CLEVR Count and TextOCR to verify the Count and OCR VFLs, respectively, under zero-shot, 1-shot, 4-shot, and 8-shot settings. Notably, the 8-shot setup exceeds 10,000 tokens, approaching Qwen-VL’s pretraining context length. CLEVR Count requires models to answer questions like “How many red objects are there in the scene?” by learning from in-context examples. TextOCR evaluates OCR ability by asking the model to output text from red-highlighted regions in images. Zero-shot results are obtained via direct text prompting without any examples.
>
> | **CLEVR Count** | zero-shot | 1-shot | 4-shot | 8-shot |
> | --- | --- | --- | --- | --- |
> | Qwen-2.5-VL | 55.5 | 60.0 | 61.5 | 59.0 |
> | Drop-OCR-VFL(20-28) | 55.0 | 60.5 | 60.5 | 59.0 |
> | Drop-Count-VFL(12-28) | 33.5 | 31.5 | 25.5 | 22.0 |
> |  |  |  |  |  |
> | **TextOCR** | zero-shot | 1-shot | 4-shot | 8-shot |
> | Qwen-2.5-VL | 46.5 | 48.0 | 47.5 | 49.0 |
> | Drop-OCR-VFL(20-28) | 12.0 | 11.5 | 9.0 | 5.5 |
> - **VFLs persist under both in-context and long-context settings.** As shown above, VFLs remain active in the in-context learning setup, and their layer-wise behavior is consistent with that observed in the zero-shot setting. Moreover, under the long-context condition (e.g., 8-shot), VFLs continue to function as expected, further confirming their stability across different prompting regimes.
> - **VFLs activate whenever a visual function is needed, not just during answer generation.** We further tested dropping only the vision tokens in the context while keeping those in the question. The performance drop is same as dropping all visual tokens, suggesting that both **task understanding** and **answering** rely on the same function layer. This confirms that **in-context prompts** and **long-context inputs** do not change the functionality or layer position of VFLs.
>
> > ***W3. Application in CoT and GRPO:***
> >
>
> We sincerely appreciate the reviewer’s insightful suggestion, which closely resonates with our own perspective.
>
> - **VFL in CoT.** During **Chain-of-Thought reasoning**, when a generated token demands a specific visual skill (e.g., OCR), the same **OCR-VFL** is activated as expected (As discussed in W2). We further observed this behavior in **math reasoning tasks**. Using the RL-trained version of Qwen2.5-vl-7B with enhanced reasoning capabilities, we conducted case analyses. When the reasoning process required a specific vision function, we fixed the random seed, reverted, and dropped the corresponding vision function layer identified earlier. This caused the model to lose that specific capability, demonstrating that **VFL consistency persists not only during chain-of-thought (CoT) generation but also remains preserved in the RL-enhanced base model**.
> - **VFL in GRPO.** We agree that applying VFLs in GRPO is a very promising direction. Current GRPO rewards focus on format and accuracy, lacking supervision or interpretability over intermediate reasoning steps (e.g., within `<\think>`). With our layer-wise localization of vision functions, the model could autonomously decide which function to invoke next (e.g., `<\OCR>`, `<\Count>`, `<\Factual>`), skipping irrelevant layers to save computation. Moreover, our method could help verify if the model truly activates the intended function, improving interpretability. However, we consider this beyond the scope of the current work but welcome further discussion and look forward to your response.
>
> > ***Typos:***
> >
>
> We will correct the typos and incorporate the above content into the revised version of the paper. Once again, we sincerely thank you for providing such valuable and constructive suggestions.
>
> [1] Not All Layers of LLMs Are Necessary During Inference. 2024
>
> [2] VL-ICL Bench: The Devil in the Details of Multimodal In-Context Learning ICLR 2025

---

> ### Author Response · Authors · 2025-08-04
>
> Thank you for acknowledging our response. We hope our additional experiments—including demonstrating that VFL remains effective on high-dimensional vision-language tasks such as A-OKVQA, in-context learning, and long-context scenarios, as well as its potential in CoT reasoning and GRPO—have sufficiently addressed your concerns.
>
> We welcome any further questions or comments during the discussion period and will do our best to respond promptly. Thank you again!

---

### Note · Authors · 2025-08-13

Dear AC and Reviewers,

We appreciate the reviewers’ recognition of our contributions, summarized as follows:

- **A Fresh Perspective and Novel Insights**: We are the **first** to systematically discovery and explore the phenomenon of **Vision Function Layers** (VFLs) within MLLMs decoder layers. Reviewers (TCaw-S3 and 5N5o-S3) recognize the validation of VFLs across 4 task types on 10 benchmarks. Meanwhile, **all reviewers** (5N5o-S1, Sn1h-S2, TCaw-S1, and 5Arb-S1) appreciate that VLFs deepens the understanding of inner workings of MLLMs.
- **A Novel Method to Target VFLs**: We design a novel method capable of precisely identifying and localizing any type of VFL, and we demonstrate 6 distinct VFL types in the paper and rebuttal. We are grateful that multiple reviewers (5N5o, Sn1h, and 5Arb) **recognize the novelty and effectiveness** of our proposed pipeline and analytical methods.
- **Two Powerful Methods Using VFLs**: We propose two methods that leverage our VFL discovery: **VFL-LoRA** and **VFL-Select**. **All reviewers** (5N5o-S4, Sn1h, TCaw-S2, and 5Arb-S2) recognize these as **powerful downstream applications** that improve MLLM performance, even **outperforming specialized methods**.
- **Clear Writing:** Reviewers (5N5o-S2 and Sn1h-S1) commend the paper for being clear, comprehensive, supporting ease of understanding.

In the rebuttal and discussion phases, we address all concerns with the following key improvements:

- We **greatly expand our evaluation** to show that **VFLs are a widespread phenomenon,** existing in more complex situations such as **in-context prompting, long-context understanding, and even videoLLMs**. We are grateful that this significant expansion is acknowledged by Reviewers (5N5o and 5Arb).
- We confirm that **VFLs appear in a wide variety of modern architecture**s, including **encoder-free, multi-encoder, and MoE-based MLLMs,** proving VFLs are **not limited to a specific model design**. Reviewer 5Arb **recognize the importance** of this verification.
- We demonstrate that our method for targeting VFLs **can identify any type of visual function of interest,** highlighting its generalizability. Please see our responses for extension details (Reviewer Sn1h) and validation (Reviewers 5N5o, TCaw).
- We demonstrate the existence and real-world applicability of VFLs through evaluation on **12** general-purpose benchmarks across a **variety of MLLMs** (see our response to Reviewer **TCaw**).

Sincerely,

The Authors

---

### Decision · Program_Chairs · 2025-09-17

**Decision:**

Accept (poster)

**Comment:**

This paper introduces Vision Function Layers (VFLs) in MLLMs, showing that core visual functions (e.g., recognition, counting, OCR, grounding) are localized in narrow blocks of layers. Using vision token dropping and swapping, the authors provide a causal framework for probing visual functions, and demonstrate practical applications through VFL-LoRA (efficient fine-tuning) and VFL-Select (data selection).

Reviewers initially raised concerns about limited scope, technical novelty, and applicability to complex tasks. However, the rebuttal by the authors added substantial new experiments, including A-OKVQA, in-context learning, CoT reasoning, video benchmarks, and diverse architectures (multi-encoder, encoder-free, MoE). These additional inputs demonstrated the robustness and generality of the findings, and convinced the reviewers on the significance of this submission.

The ACs like the findings of this work regarding the vision function layer and the application of this discovery for improve the parameter-efficient fine-tuning and data selection for model training. While the work is more analytical than architectural, its clarity and novelty make it a valuable and timely contribution to MLLM interpretability and efficiency. With reviewers converging to borderline accept or higher (one raising to full accept), the ACs recommend acceptance.